# Independent Component Discovery in Temporal Count Data

Alexandre Chaussard [1]   Anna Bonnet [1]   Sylvain Le Corff [1]

## Abstract

Advances in data collection are producing growing volumes of temporal count observations, making adapted modeling increasingly necessary. In this work, we introduce a generative framework for independent component analysis of temporal count data, combining regime-adaptive dynamics with Poisson log-normal emissions. The model identifies disentangled components with regime-dependent contributions, enabling representation learning and perturbations analysis. Notably, we establish the identifiability of the model, supporting principled interpretation. To learn the parameters, we propose an efficient amortized variational inference procedure. Experiments on simulated data evaluate recovery of the mixing function and latent sources across diverse settings, while real-world applications to gut microbiome and climate datasets reveal co-variation patterns and regime shifts consistent with domain-specific knowledge.

## 1. Introduction

Multivariate temporal count data arise in many application areas, notably healthcare, finance or climate science, where both interpretability and faithful characterization of perturbations and regimes (e.g. infections, trends, seasons) are increasingly central (Lyu et al., 2023; Bacry et al., 2015; Koh et al., 2023). Despite their prevalence, principled modeling for temporal count data remains challenging: observations are non-negative, discrete, frequently overdispersed, and often exhibit heavy-tailed variability. While a substantial body of work focuses on forecasting models for temporal count data (see Davis et al. (2021) and the references therein), these approaches provide more limited tools for theoretically grounded analyses of latent structures, dimension reduction, and the extraction of interpretable mechanisms.

[1]CNRS, Laboratoire de Probabilités, Statistique et Modélisation, LPSM, Sorbonne Université, F-75005 Paris, France. Correspondence to: Alexandre Chaussard <alexandre.chaussard@sorbonne-universite.fr>.

*Proceedings of the 43rd International Conference on Machine Learning*, Seoul, South Korea. PMLR 306, 2026. Copyright 2026 by the author(s).

In this context, Independent Component Analysis (ICA) provides an attractive framework for interpretable representation learning by expressing observed signals as mixed independent latent sources (Hyvärinen, 2013). In temporal settings with linear mixing, the dependency structure can be leveraged to characterize the mixing map and latent sources (Belouchrani et al., 2002), supporting principled interpretation. Recent works then generalize identification results to structured ICA and nonlinear mixing models (Morioka et al., 2021; Hälvä et al., 2024), improving on model expressivity. While these developments enable principled downstream analysis when observations are real-valued and corrupted by additive noise, their framework is not well-suited for temporal count data, where discreteness and heteroskedastic noise can lead to misspecification or unstable inference, highlighting the need for ICA models tailored to count data.

In contrast, count data analysis typically relies on observation models that explicitly capture their statistical properties, such as negative binomial likelihoods (Hilbe, 2011) or Poisson mixed models. Among these, Poisson log-normal (PLN) generative models (Aitchison, 1982) are particularly compelling, as their latent Gaussian variables induce flexible dependencies across dimensions while preserving a well-grounded probabilistic interpretation (Chiquet et al., 2021). Yet, while Poisson mixed models are showing promises in temporal count data applications, their potential for representation learning and ICA remain underexplored.

In this work, we propose a generative framework for ICA of temporal count data using PLN models. To accommodate exogenous perturbations and regime changes in dynamics, we couple the PLN observation model with a switching linear dynamical system (Ghahramani & Hinton, 2000), enabling latent regime segmentation and globally nonlinear dynamics. Leveraging the model's structure, we establish identifiability guarantees on the latent components under mild assumptions, supporting principled downstream interpretation. To learn the model parameters, we derive an efficient amortized variational inference procedure (Blei et al., 2017) that exploits conditional structure and uses a deep variational parameterization designed to scale to multivariate time series. To complement our identifiability results, we study several simulation regimes and demonstrate partial recovery of the mixing map in finite-sample setting. We then demonstrate the empirical relevance of the model on

two real-world datasets: a gut microbiome study (Bucci et al., 2016), where inferred regimes and components align with clinical perturbations and medically relevant microbial interactions; and a severe weather hazard dataset (NOAA et al., 1996), where inferred regimes track seasonal structure while components recover known meteorological patterns. We also report an auxiliary forecasting benchmark against standard baselines as a quantitative evaluation of the model.

Our methodological contributions are the following: (i) a generative model for ICA of temporal count data with regime switching for perturbation-aware analysis, (ii) identifiability guarantees for the mixing map up to standard ICA indeterminacies, and (iii) a structured amortized variational inference procedure to learn the model parameters.

**Notations.** Consider a dataset $\mathcal{D}$ of $N$ temporal count data of length $T$. We write $v_{1:T}^{1:K}$ to represent a multivariate time series of length $T$ with $K$ features per time step, i.e. for each $x \in \mathcal{D}$, we write $x = x_{1:T}^{1:K} \in \mathbb{N}^{T \times K}$. For all $1 \leq t \leq T$, $1 \leq i \leq K$, the $i$-th feature of $v_t$ is written $v_t^{(i)}$. For all sequences $z = (z_j)_{1 \leq j \leq K} \in \mathbb{R}^K$, we write $x \sim \mathcal{P}(\exp z)$ when $(x_j)_{1 \leq j \leq K}$ are independent with $x_j \sim \mathcal{P}(\exp z_j)$. We denote functions under the model by $p_{\boldsymbol{\theta}}$ with parameters $\boldsymbol{\theta}$, and $q_{\boldsymbol{\varphi}}$ the variational proxy with parameters $\boldsymbol{\varphi}$.

## 2. Background and related works

**Independent Component Analysis.** A central goal in data analysis is to establish interpretable representations of high-dimensional observations, which are often correlated and difficult to analyze directly. ICA addresses this by assuming that observations are generated from *component-wise independent* latent sources of possibly smaller dimension, mixed through an unknown *mixing* function (Hyvärinen, 2013). Denoting the sources by $s = s_{1:T}^{1:d} \in \mathbb{R}^{T \times d}$ and a parametric mixing map $f_{\boldsymbol{\theta}} : \mathbb{R}^d \rightarrow \mathbb{R}^K$, the observations are given by $x_t = f_{\boldsymbol{\theta}}(s_t), 1 \leq t \leq T$. To explicitly account for observation noise and enable principled inference, probabilistic ICA specifies a generative model via a prior $p_{\boldsymbol{\theta}}(s_{1:T})$ and an observation model $p_{\boldsymbol{\theta}}(x_{1:T} \mid s_{1:T})$, enabling likelihood-based learning and uncertainty quantification (Hälvä et al., 2024; Hyvarinen & Morioka, 2016). A key challenge in ICA is the characterization of the mixing map $f_{\boldsymbol{\theta}}$. In linear ICA with additive noise, identifiability holds under some assumptions, typically excluding more than one Gaussian source (Hyvärinen & Oja, 2000), or by assuming joint diagonalization of auto-correlation matrices in time series (Belouchrani et al., 2002); both yielding mixing identification up to column-wise permutation and rescaling. Beyond the linear case, nonlinear ICA is in general not identifiable without additional structure, and temporal information is one principled route to restore identifiability (Hyvarinen & Morioka, 2016). Recent work for-

malizes structured nonlinear ICA models and establishes identifiability even with additive noise of unknown distribution (Hälvä et al., 2024). However, these formulations do not directly extend to count data, whose discreteness, non-negativity, heteroskedasticity and overdispersion make continuous additive-noise models ill-suited. In this work, we adapt the structured ICA viewpoint of Hälvä et al. (2024) to count data by using a Poisson-based emission model, treating observation noise through composition in the likelihood, and enabling identifiability under mild assumptions.

**Structured count data.** Count data are commonly found in many applications, and pose distinct challenges due to discreteness, non-negativity, and frequent overdispersion, limiting the applicability of standard modeling tools (Cameron & Trivedi, 2013). Among count-specific approaches, Poisson Log-Normal (PLN) models provide a principled framework for capturing dependencies via latent Gaussian log-intensities, enabling downstream tasks such as network inference and principle component analysis (Chiquet et al., 2019; 2021). Recent extensions to structured settings further highlight the flexibility of the framework and its practical interest (Batardière et al., 2025; Chaussard et al., 2025b; Qian et al., 2024); yet, PLN models tailored to temporal structures remain largely underexplored. Meanwhile, several models have been proposed for temporal count data, including integer auto-regressive models (Jin-Guan & Yuan, 1991), binary series (Cui & Lund, 2009; Livsey et al., 2018), and negative binomial or Poisson regression (Davis & Wu, 2009; Fokianos et al., 2009); however, these methods are mostly forecasting-oriented and provide limited tools for interpretability-driven analysis. Extensions of Poisson mixed models have then been developed, notably switching linear dynamical models for regime segmentation in neuroscience applications (Zoltowski et al., 2020; Glaser et al., 2020). In ecology, explainable mechanistic models such as generalized Lotka-Volterra dynamics are also widely employed to describe species interactions over time (Stein et al., 2013; Gibson & Gerber, 2018; Gibson et al., 2025), but they are tied to specific applications and often lack identifiability guarantees supporting interactions analysis. The generative framework introduced in this paper aims at addressing these limitations by extending PLN modeling to ICA of temporal count data while retaining a tractable structure for inference, and theoretical guarantees supporting downstream analysis.

## 3. ICA for temporal count data

### 3.1. Generative Model

**General formulation.** The proposed PLN-ICA framework models temporal count observations as conditionally independent Poisson variables whose intensities are

obtained by mixing component-wise independent latent sources. Formally, letting $f_{\boldsymbol{\theta}} : \mathbb{R}^d \to \mathbb{R}^K$, $K \geq d > 0$, we assume that (i) the latent sources $(s_{1:T}^{(i)})_{1 \leq i \leq d}$ are mutually independent, (ii) $z_t = f_{\boldsymbol{\theta}}(s_t)$ defines the mixed log-intensities at time $t \leq T$, and (iii) conditional on $s_{1:T}$, the observations $x = x_{1:T}^{1:K}$ are independent with $x_t^{(k)} \sim \mathcal{P}(\exp z_t^{(k)})$ for $1 \leq k \leq K$. This construction inherits key advantages of PLN modeling: Poisson emissions enforce non-negativity, while latent log-normal intensities induce overdispersion and heavy-tail modeling. From a representation-learning ICA perspective, the sources $s$ provide a low-dimensional, disentangled description of multivariate count trajectories. Moreover, observation noise is modeled directly through Poisson sampling, yielding a non-additive, non-Gaussian model. As such, the PLN-ICA framework provides a structured ICA model tailored to temporal counts, departing from prior identifiable ICA formulations developed for real-valued additive-noise observations.

**Auto-Regressive PLN-ICA.** The PLN-ICA framework encapsulates a large variety of models depending on the prior over $s_{1:T}$ and the choice of mixing function $f_{\boldsymbol{\theta}}$. In this work, we focus on a linear mixing model combined with *switching* linear Gaussian dynamics, which we refer to as Auto-Regressive PLN-ICA (ARPLN-ICA, see Figure 1):

- **Component-wise switching**: For all $1 \leq i \leq d$, $u^{(i)}$ is a discrete Markov chain with initial distribution $\pi^{(i)}$ and transition matrix $A^{(i)} = (a_{k\ell}^{(i)})_{1 \leq k, \ell \leq C}$.

- **Regime dependent sources**: For all $1 \leq i \leq d$, conditionally on $\{u_{t+1}^{(i)} = k, \ s_t\}$,

$$s_{t+1}^{(i)} \sim \mathcal{N}\big(B_k^{(i)} s_t^{(i)} + b_k^{(i)}, \psi_k^{(i)}\big) , \qquad (1)$$

with $B_k^{(i)}, b_k^{(i)} \in \mathbb{R}$, $\psi_k^{(i)} \in \mathbb{R}^+$, and $s_1$ following a multivariate Gaussian conditionally on $\{u_1 = k\}$ with mean $\bar{b}_k$ and diagonal covariance $\bar{\psi}_k$.

- **PLN-ICA linear mixing**: Given a linear mixing $\Gamma \in \mathbb{R}^{K \times d}$, for all $1 \leq t \leq T$, conditionally on $s_t$,

$$x_t \sim \mathcal{P}\big(\exp(\Gamma s_t)\big) . \qquad (2)$$

The PLN emission enforces a statistical framework suited to counts, converting the latent AR variability into discrete noise, while regime switching further accommodates non-stationary and nonlinearity commonly found in applications. The latent sources are then linearly mixed into multivariate log-intensities via $\Gamma$, yielding an interpretable decomposition of temporal dynamics. In the next section, we establish the identifiability class of $\Gamma$ under mild conditions, providing guidelines on principled downstream interpretation.

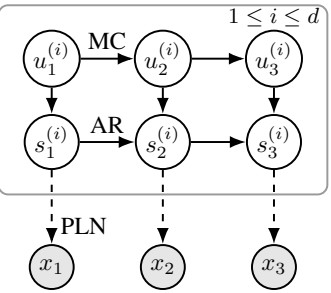

*Figure 1.* ARPLN-ICA graphical dependencies representation. MC indicates the discrete Markov chain on the latent switching labels modeling regimes, AR the Gaussian auto-regressive prior on the sources conditional to the latent regimes, and PLN the Poisson Log-Normal linear mixing emission of counts at time $t$.

### 3.2. Identifiability results

Identifiability is a central property of ICA models, with both practical and theoretical implications. Formally, a model is identifiable if its parameters are uniquely determined by the distribution of the observations. From a theoretical standpoint, lack of identifiability undermines meaningful interpretation, for instance when recovering an interaction network or estimating independent components. In practice, non-identifiability can lead to ill-posed estimation due to invariance classes and may even degrade generalization (D'Amour et al., 2022). For these reasons, we explore the identifiability of PLN-ICA models in the following results. All proofs are postponed to Appendix A.

**Proposition 3.1.** *Let* $(x, s) = (x_t, s_t)_{1 \leq t \leq T}$ *(resp.* $(\tilde{x}, \tilde{s})$*) such that* $x \in \mathbb{N}^{T \times K}$ *and* $s \in \mathbb{R}^{T \times d}$ *with* $K \geq d > 0$. *Let* $f$ *(resp.* $\tilde{f}$*) a mapping from* $\mathbb{R}^d$ *to* $\mathbb{R}^K$, *and assume that conditionally on* $s_t$, $x_t \sim \mathcal{P}(\exp f(s_t))$ *(resp.* $\tilde{x}_t \sim \mathcal{P}(\exp \tilde{f}(\tilde{s}_t))$*). Then, if* $x$ *and* $\tilde{x}$ *have the same distribution,* $(f(s_t))_{1 \leq t \leq T}$ *and* $(\tilde{f}(\tilde{s}_t))_{1 \leq t \leq T}$ *have the same distribution.*

Proposition 3.1 establishes that the identifiability class of PLN-ICA models coincides with the invariance class of the corresponding noiseless ICA problem on the log-intensities. This key modeling advantage of PLN models notably allows to alleviate restrictive assumptions found in noise-additive structured ICA (Hälvä et al., 2024), allowing to compose any identifiable ICA framework with PLN distributions while preserving global identifiability. Notably, Corollary 3.2 shows that the remaining invariances of ARPLN-ICA reduce to standard permutation and rescaling of the columns of the mixing for switching linear ICA. Likewise, further nonlinear PLN-ICA identifiability results can be derived for more complex ICA modeling, for instance by relying on Theorem 2 of Hälvä et al. (2024).

**Corollary 3.2.** *Let* $(x, s, u) = (x_t, s_t, u_t)_{1 \leq t \leq T}$ *(resp.* $(\tilde{x}, \tilde{s}, \tilde{u})$*) such that* $x \in \mathbb{N}^{T \times K}$, $s \in \mathbb{R}^{T \times d}$, $u \in \mathbb{R}^{T \times C}$ *(resp.* $\tilde{x}, \tilde{s}, \tilde{u}$*) with* $K \geq d > 0$ *and* $C > 0$. *Assume* $(x, s, u)$, $(\tilde{x}, \tilde{s}, \tilde{u})$ *are distributed according to dis-*

tinct ARPLN-ICA models, with $\Gamma \in \mathbb{R}^{K \times d}$ (resp. $\tilde{\Gamma}$) full rank. Assume $\Sigma_1 = \mathbb{C}\mathrm{ov}(s_1, s_1)$ has positive diagonal entries, and that there exist $t_0 \leq T$ and $\ell_0 < t_0$ such that $\Sigma_1^{-1/2} \mathbb{C}\mathrm{ov}(s_{t_0}, s_{t_0 - \ell_0}) \Sigma_1^{-1/2}$ diagonal entries are non-zero pairwise distinct. Then, if $x$ and $\tilde{x}$ have the same distribution, $\Gamma$ and $\tilde{\Gamma}$ are equal up to column-wise permutation and non-zero rescaling.

The assumptions underlying Corollary 3.2 are generically satisfied, as the indeterminacy of $\Gamma$ arises only on a Lebesgue measure zero subset of parameter configurations. In fact, the conditions that the initial covariance of the sources is positive definite, and that the whitened lag-covariance has nonzero pairwise distinct diagonal entries can be interpreted as mild non-degeneracy assumptions. Notably, Appendix A.3 shows that in the homogeneous case $C = 1$, the AR coefficient matrix $B$ enforces the required whitening condition whenever it has pairwise distinct non-zero diagonal entries, which only fails on a measure-zero set under any absolutely continuous prior over $B$.

These limited indeterminacies on $\Gamma$ enable a principled characterization of interactions between the latent components and the counted entities, where the relationship is given by the log-intensity loadings of entity $k \leq K$ at time $t \geq 1$ as

$$z_t^{(k)} = \sum_{i=1}^{d} \Gamma_i^{(k)} s_t^{(i)} . \tag{3}$$

Since the mixing matrix $\Gamma$ is identifiable only up to column-wise permutation and rescaling, entity-component relationships are best described component-wise, in terms of relative loading magnitudes, and using a fixed ordering of components. To handle the rescaling ambiguity, one may fix the column norms of $\Gamma$ without loss of generality, since any column-wise rescaling can be absorbed by an inverse rescaling of the corresponding latent component, together with the induced reparameterization of the latent AR dynamics.

### 3.3. Variational inference and learning

As with standard PLN models, ARPLN-ICA yields an intractable observed likelihood $p_{\boldsymbol{\theta}}(x)$, and the posterior $p_{\boldsymbol{\theta}}(z \mid x)$ does not admit a closed form. While Monte Carlo estimators can approximate the likelihood and its gradients, these methods entail practical challenges and substantial computational cost. Conversely, variational inference has shown efficient training and fast sampling upsides (Blei et al., 2017), making them favorable alternatives in practice. Variational training consists in approximating the posterior with a tractable proxy $q_{\boldsymbol{\varphi}}(z \mid x)$, then maximizing a surrogate objective called evidence lower bound (ELBO):

$$\mathcal{L}(x; \boldsymbol{\theta}, \boldsymbol{\varphi}) = \mathbb{E}_{q_{\boldsymbol{\varphi}}(\cdot \mid x)}[\log p_{\boldsymbol{\theta}}(x, z) - \log q_{\boldsymbol{\varphi}}(z \mid x)] . \tag{4}$$

**Variational proxy.** To learn the parameters of ARPLN-ICA, we consider a structured variational family that factorizes across latent coordinates and separates the continuous sources from the discrete regimes given the observed counts,

$$q_{\boldsymbol{\varphi}}(s, u \mid x) = \prod_{i=1}^{d} q_{\boldsymbol{\varphi}}\big(s_{1:T}^{(i)} \mid x\big) \, q_{\boldsymbol{\varphi}}\big(u_{1:T}^{(i)} \mid x\big) . \tag{5}$$

This approximation assumes coordinate and source-label conditional independence, while allowing arbitrary temporal dependencies within each factor, making it further structured than a typical mean-field approximation (Blei et al. (2017), Appendix E.1). Motivated by the latent Markov structure of the true posterior distribution, we further parameterize each variational source factor as a linear Gaussian Markov chain,

$$q_{\boldsymbol{\varphi}}\big(s_{1:T}^{(i)} \mid x\big) = \mathcal{N}\left(s_1^{(i)}; m^{(i)}(x), \tilde{\psi}_1^{(i)}(x)\right)$$
$$\times \prod_{t=1}^{T-1} \mathcal{N}\left(s_{t+1}^{(i)}; \tilde{B}_{t+1}^{(i)}(x) \, s_t^{(i)} + \tilde{b}_{t+1}^{(i)}(x), \tilde{\psi}_{t+1}^{(i)}(x)\right), \tag{6}$$

where $m^{(i)}(\cdot), \tilde{B}_t^{(i)}(\cdot)$, and $\tilde{b}_t^{(i)}(\cdot)$ are functions of the observations mapping to $\mathbb{R}$ and $\tilde{\psi}_t^{(i)}(\cdot)$ maps to $\mathbb{R}_{>0}$, with parameters $\boldsymbol{\varphi}$. This construction retains the Markov structure of the posterior while enabling efficient inference via marginal recursions (see Appendix B). For the discrete regimes, selecting the variational proxy in the exponential family yields an optimal closed-form distribution thanks to the model's conjugacy (Johnson et al., 2016), enabling fast and optimal inference. See Appendix C for its derivation.

**Learning procedure.** We derive a closed-form expression of the ARPLN-ICA ELBO in Appendix B thanks to tractable forward recursions of the variational distribution. The result highlights close connections with the PLN ELBO of Chiquet et al. (2019), up to additional contributions induced by the temporal Markov structure of ARPLN-ICA. We estimate the model parameters by maximizing the ELBO

---

**Algorithm 1** Stochastic VEM for ARPLN-ICA

---

**Require:** Observations $x_{1:T}$
  Initialize $\boldsymbol{\theta}^{(0)}, \Gamma^{(0)}$, and $\boldsymbol{\varphi}^{(0)}$
  **for** $h = 0, \dots, H - 1$ **do**
    **Expectation step.**
    Compute recursive marginals of $q_{\boldsymbol{\varphi}^{(h)}}(s \mid x)$
    $q_{\boldsymbol{\varphi}^{(h)}}(u \mid x) \leftarrow \mathrm{CAVI}\big(\boldsymbol{\theta}^{(h)}, \Gamma^{(h)}, \boldsymbol{\varphi}^{(h)}\big)$

    **Maximization step.**
    $\big[\Gamma^{(h+1)}, \boldsymbol{\varphi}^{(h+1)}\big] \leftarrow \mathrm{SGD}\big(x; \boldsymbol{\theta}^{(h)}, \Gamma^{(h)}, \boldsymbol{\varphi}^{(h)}\big)$
    $\boldsymbol{\theta}^{(h+1)} \leftarrow \mathrm{CLOSEDFORM}\big(x; \boldsymbol{\theta}^{(h)}, \boldsymbol{\varphi}^{(h+1)}\big)$
  **end for**
  **return** $(\boldsymbol{\theta}^{(H)}, \Gamma^{(H)}, \boldsymbol{\varphi}^{(H)})$

---

with a stochastic variational EM (VEM) procedure that alternates between (i) computing the variational factors and evaluating the ELBO, and (ii) updating the generative parameters, as summarized in Algorithm 1. The variational update of the latent labels leverages the variational proxy of the sources to compute $q_{\varphi}(u \mid x)$ via coordinate-ascent variational inference (CAVI; see Appendix C). Conditional on these variational quantities, most parameters in $\boldsymbol{\theta}$ admit closed-form optima (see Appendix D). While the mixing matrix $\Gamma$ does not yield a closed-form update, we optimize it with a stochastic gradient step, jointly with the neural networks parameterizing the variational sources distribution.

**Model implementation.** In practice, specifying the variational proxy requires to map the count sequence $x_{1:T}$ to the parameters of the variational factors in Eq. (6). However, the time-dependent dimension of the observation can make the learning procedure challenging. To capture temporal dependencies in a way that is consistent with the Markov structure of the posterior, we encode $x_{1:T}$ into a sequence of fixed-dimensional representations $\tilde{x}_t \in \mathbb{R}^{d_e}$ using a gated recurrent unit (GRU), similar to Chaussard et al. (2025a). For each time step $t$, the variational parameters are then predicted from $(x_t, \tilde{x}_t)$, mimicking a residual connection to preserve local information. This amortized parameterization illustrated in Figure 10 can scale to any temporal input, enabling fast inference while leveraging the flexibility of neural networks to represent rich dependencies. Furthermore, while Corollary 3.2 establishes the identifiability of the model, the mixing invariances can hinder optimization through underspecification effects (D'Amour et al., 2022). To address the scaling indeterminacy, we propose to restrict the equivalence class to signed permutations by enforcing a column-wise normalization of the mixing matrix during training, that is, each column of Gamma has unit $\ell_2$-norm during training. This constraint is lightweight and compatible with stochastic optimization, and does not restrict the set of representable ARPLN-ICA models, as the overall scale of each source can still be expressed through the unconstrained mean and variance parameters of the latent linear Gaussian processes. To facilitate use and reproducibility, we provide an efficient PyTorch implementation of our model with GPU support, available on GitHub[1], together with Python scripts allowing to reproduce our experiments.

### 3.4. Fixed effects and measurement noise modeling

In many count-data applications, natural discrepancies and systematic exogenous factors such as sampling effort (offset) can partially explain variability over time. Following Chiquet et al. (2019), we incorporate these effects additively in the log-intensity of the emission model. Let $\eta \in \mathbb{R}^K$ denote entity-specific fixed effects (baseline log-abundances) and

---

[1] github.com/AlexandreChaussard/PLNICA

let $o_t \in \mathbb{R}$ be a known sampling-effort offset at time $t$ (e.g., library size, sequencing depth, exposure). Conditionally on $s_t$, we model the observed counts as

$$x_t \sim \mathcal{P}(\exp\{\Gamma s_t + o_t \mathbf{1}_K + \eta\}) , \qquad (7)$$

where $\mathbf{1}_K$ denotes the $K$-dimensional vector of ones. Notably, as these deterministic effects enter additively in the log-intensity, they cancel in the centered mixed process, maintaining the validity of the proof of Corollary 3.2. Inference for the fixed-effects coefficients $\eta$ is conducted jointly with $\Gamma$ through stochastic optimization.

## 4. Simulations

**Mixing recovery simulation setup.** Corollary 3.2 establishes the identifiability of the mixing $\Gamma$ under mild assumptions. To support this asymptotic result in a finite-sample regime, we study the empirical recovery of $\Gamma$ under three simulated settings. The first two settings evaluate the effect of column colinearity on recovery: a *moderate-coherence* regime where columns are weakly colinear, and a *high-coherence* regime where columns are strongly colinear. Then, we consider a moderate-coherence, *low-excitation* regime in which the observed counts exhibit a high proportion of zeros. Across all scenarios, we fix $K = 12$, $T = 20$, $C = 1$, $d = 5$, and sample $n = 150$ trajectories from an oracle model. We then train 10 ARPLN-ICA models with same latent dimension $d$ and fixed variational architecture, with parameters initialized with different random initializations. Fit to the oracle distribution is quantified using the sliced Wasserstein distance, while recovery of $\Gamma$ is assessed column-wise via cosine similarity after column-wise normalization and optimal matching to mitigate signed permutation invariance (see Appendix E.4). We additionally report mixing recovery obtained with two widely used linear ICA methods: UWEDGEICA, a second-order identification approach for temporal ICA (Pfister et al., 2019), and PICARD, a robust general-purpose ICA algorithm (Ablin et al., 2018). For comparability, both baselines are fit on log-counts, which under ARPLN-ICA model yield an approximately linear ICA task with mixing $\Gamma$ (see Appendix E.2). Since these methods only estimate $\Gamma$, they do not define generative models, and therefore we do not report distributional fit to the oracle. Finally, we compare our structured variational inference procedure (AR, see Eq.(6)) to a classical mean field (MF) approximation (Blei et al., 2017), in which $q_{\varphi}(s_{1:T}^{(i)} \mid x)$ factorizes along time (see Appendix E.1). Additional details on the simulation design, evaluation metrics and compared methods are provided in Appendix E.

**Empirical recovery of diverse mixing.** Figure 2 summarizes recovery performance across the three simulation settings. Under both moderate and high coherence, the proposed structured AR inference reaches partial component-

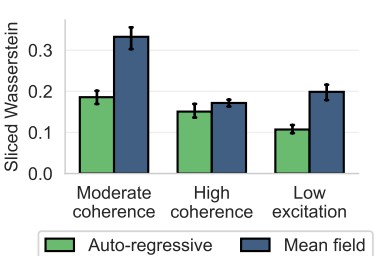 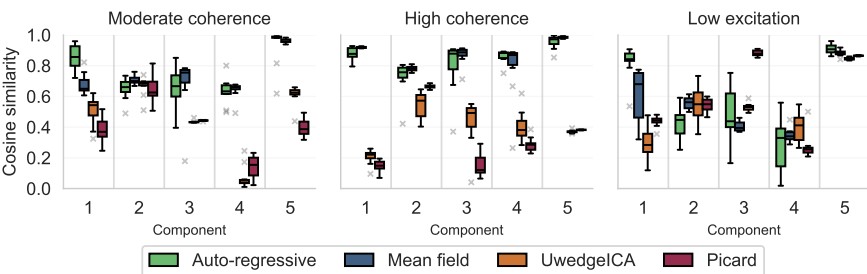

*Figure 2.* Generative distribution quality and mixing recovery in simulation scenarios with ARPLN-ICA, comparing Auto-Regressive (AR) and Mean-Field (MF) inference of the mixing in ARPLN-ICA, and linear ICA mixing estimation algorithms. (Left) Sliced Wasserstein evaluates the distance to the baseline distribution with the fitted model (mean with 95% CI; lower is better). (Right) Cosine similarity evaluates the absolute scalar product between the columns of the baseline mixing and the fitted mixing (1: recovery, 0: orthogonality).

wise recovery with the oracle mixing matrix, with average cosine similarities of $74.6\%$ and $84.5\%$, respectively. Despite the reduced separation forced in *high coherence* setting, the procedure still reliably discriminates between closely aligned columns. In the low-excitation regime, recovery becomes more heterogeneous across components, with the less discriminative components relative to the log-intensity contribution being the hardest to identify (see Figure 9, Appendix E.3). Overall, these results suggest that recovery is primarily driven by the magnitude of the latent log-intensities: when components induce only subtle variations in the counts, finite-sample identification is more challenging. Additionally, UWEDGEICA and PICARD generally underperform, with respective average recovery of $46\%$ and $44\%$ across scenarios, indicating that linear ICA on log-counts is sensitive to observation-model mismatch: the surrogate $\log$-counts only approximately fits the additive linear ICA framework while introducing heteroskedasticity, which degrades identification even when leveraging temporal structure (UWEDGEICA) or robust estimation schemes (PICARD), showing the need for count-specific ICA models.

Beyond parameter recovery, the structured variational family yields clear improvement in fit to the oracle generative model, as measured by the sliced Wasserstein, compared to a mean-field approximation. In contrast, both inference procedures achieve comparable levels of recovery of $\Gamma$ across scenarios. Taken together, these findings support the use of structured variational approximations to better capture temporal dependencies and improve distributional fit.

**Effect of the number of training trajectories.** We evaluate the dependence of mixing recovery on sample size by varying the number of training trajectories $n$ in the moderate-coherence regime. As shown in Figure 3, the recovery of $\Gamma$ improves consistently with $n$, with diminishing returns beyond $n \approx 80$ in this setting. The resulting plateau in partial recovery may reflect practical limitations of amortized variational inference, including the variational approximation gap, amortization-induced bias, and optimization effects (Shu et al., 2018). Appendix E.5 provides an additional timing evaluation in the same setting, illustrating the dependence of wall-clock runtime on $n$ and $T$.

## 5. Illustration on microbiome data

### 5.1. Microbiome dataset and study design

**Gnotobiotic mice dataset.** Microbiome studies provide a quantitative view of microbial communities and are increasingly used to understand disease mechanisms and guide treatment conception (Schmidt et al., 2018; Asnicar et al., 2024). Genome sequencing using high-throughput metagenomic pipelines allows to explore the composition of the gut microbiome, forming abundance profiles over microbial taxa living in the gut. In this work, we consider the *in vivo* longitudinal dataset of Bucci et al. (2016), which measures gut microbiome dynamics in response to *Clostridium difficile* infection. The study tracks $n = 5$ mice for 56 days, with controlled gut microbial community (gnotobiotic). Samples are collected approximately every two days, yielding $T = 26$ microbiome count observations per mouse. Two controlled perturbations induce distinct dynamical regimes: mice are first colonized with $K = 12$ human commensal species (index $t = 0$) and allowed 28 days of rest to reach a stable community; at day 28 (index $t = 13$),

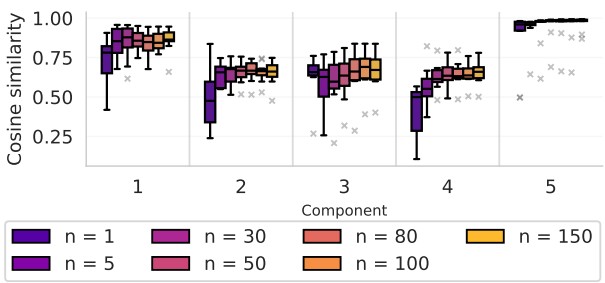

*Figure 3.* Mixing components recovery with varying training size $n$ using the Auto-Regressive variational approximation in the moderate coherence scenario. Recovery is assessed via cosine similarity per component over 10 training with different initializations.

all mice are infected with *C. difficile* spores and monitored for an additional 28 days. Overall, this dataset provides a challenging modeling task in low-sample size regime with dynamic shifts induced by perturbations, matching the context of our model.

**Number of ICA components.** Choosing the number of latent components $d$ is a central modeling decision in ICA. In our experiments, we set $d = 4$ to prioritize interpretability and facilitate visualization of the learned trajectories and loadings. This choice is not claimed to be optimal; rather, it reflects a practical trade-off between descriptive resolution and readability. We provide a sensitivity analysis and selection procedure for $d$ in Appendix F.4 based on reconstruction performance, which indicates stable behavior across a range of values of $d$. Notably, qualitatively similar component-level interpretations can be obtained for larger $d$, albeit with reduced visual tractability. We further assess the stability of the analysis at $d = 4$ in Appendix F.3.

**Training procedure.** In the following experiments, we set the latent dimension to $d = 4$ sources and use $C = 2$ switching regimes. We use the encoder architecture employed in the simulation study (Figure 10, Appendix E.3) and mitigate offset noise by considering logsum effects (see Appendix G.4). Given the small number of subjects, we regularize training using weight decay via AdamW (Loshchilov & Hutter, 2019). Full architectural and optimization details are provided in Appendix F.1. All evaluations are performed under leave-one-out cross-validation (LOO) across mice.

### 5.2. Unsupervised perturbation modeling

Exogenous perturbations, such as medical interventions or environmental changes, can induce abrupt shifts in gut microbiome dynamics. While most existing approaches assume that perturbation times are known (Gibson & Gerber, 2018; Gibson et al., 2025), ARPLN-ICA can leverage its switching dynamics to infer hidden regimes in an unsupervised framework. In the gnotobiotic mice study, two perturbations are documented: the initial gut colonization and the subsequent *C. difficile* infection. Accordingly, we

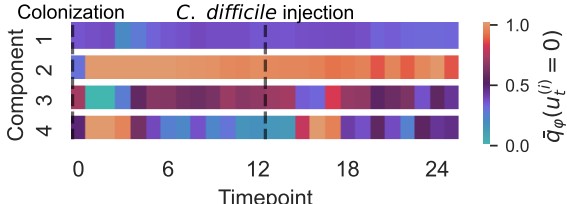

*Figure 4.* Average probability of being in latent state 0 for each component, computed on test mice across LOO CV. The probability is computed from empirical inhomogeneous derivation in Appendix F.2. Evaluation per test mouse is provided in Figure 13.

consider $C = 2$ switching states to model pre- and post-infection regimes. Figure 4 reports the posterior probability of being in regime 0 over time on test samples in the LOO CV. Following colonization at $t = 0$, all components exhibit a transient departure from their starting regime before stabilizing around $t = 5$, consistent with the establishment of a commensal community following an initial proliferation phase. After *C. difficile* infection at $t = 13$, components 3 and 4 show renewed regime switch, while the other components remain comparatively stable. These components then return to a stable regime by $t = 18$, aligning with the recovery phase reported in Bucci et al. (2016), suggesting that these components capture infection-driven mechanisms.

### 5.3. Interaction modeling

The mixing matrix $\Gamma$ defines a linear map from latent sources to the Poisson log-intensities modeling each microbial abundance. Notably, given the log-intensities in Eq. (3), the $k$-th row of $\Gamma$ summarizes how strongly taxon $k$ relies on each latent source to explain its abundance, with relatively larger absolute coefficients indicating stronger association, while the $i$-th column characterizes the taxa-level association patterns with source $i$. Since $\Gamma$ is column-normalized during inference, Corollary 3.2 implies that it is identifiable only up to a signed permutation; as a result, taxa-source associations should be interpreted in terms of *co-variation* rather than uniquely oriented effects. To ensure meaningful comparisons across LOO-CV folds, we align all estimated mixing matrices to a reference medoid, defined as the fold-specific $\Gamma$ with the highest average pairwise similarity across folds. Additional details on medoid construction and component stability are provided in Appendix F.3.

Figure 5 reports the resulting taxa co-variation patterns for each component in the gnotobiotic experiment, together with the temporal evolution of the inferred latent sources. Focusing on component 4, the infection-related attribution from Section 5.2 is corroborated by the abrupt trend shift in Figure 5, accounting for $67\%$ of the source-wise absolute mean shift between commensal ($8 \leq t \leq 13$) and infection phases ($14 \leq t \leq 19$). Moreover, taxa loadings of component 4 indicate strong positive co-variation between *C. difficile* and taxa frequently reported as dysbiosis-associated bacteria or opportunistic pathogens of the gut microbiome, including *E. coli*, *B. fragilis*, and *C. ramosum* (Fujimoto et al., 2020; Patrick, 2022). In contrast, *C. difficile* exhibits negative co-variation with *A. muciniphila* which is often linked to anti-inflammatory profiles and metabolic health (Derrien et al., 2017). Overall, ARPLN-ICA provides an interpretable view on microbial interactions consistent with Bucci et al. (2016) analysis, with regime switching dynamics helping to localize components that exhibit distinct functional responses over time, and lower dimensional representations facilitating the interpretation of complex dynamics.

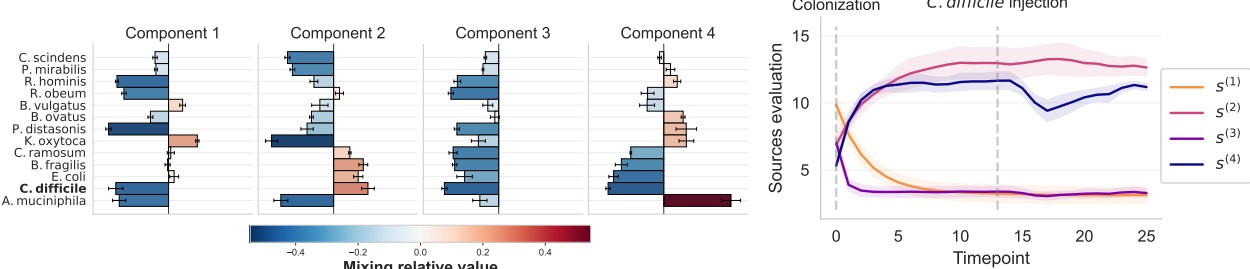

*Figure 5.* Independent components inferred by ARPLN-ICA on the gnotobiotic mice dataset. (Left) Taxa loading per component (column) of the medoid mixing across LOO CV, with standard deviation to the medoid across folds. (Right) Latent components evolution through time for test mice, aligned with the medoid mixing across LOO, with standard deviation. The fourth component displays a sharp reaction to the perturbation, consistent with Figure 4, with taxa loadings separating opportunistic pathogens from healthy and commensal microbial species. See Appendix F.3 for the computation of the medoid mixing and stability of components.

### 5.4. Evaluation on microbial forecasting

Beyond providing qualitative interpretations matching domain-specific knowledge, the interest of a dynamical model can be assessed through its ability to capture temporal structure that generalizes across unseen trajectories. Forecasting provides such an auxiliary quality-of-fit task, secondary to the representation learning objective of ICA, by comparing held-out trajectory segments with model predictions. To that end, we provide a quantitative benchmark against standard forecasting methods used in microbiome analysis as well as temporal ICA baselines. This evaluation requires an online inference procedure and an appropriate sampling scheme which are detailed in Appendix G.

**Evaluation design.** We compare ARPLN-ICA forecasting against a set of complementary baselines typically encountered in microbiome analysis, covering both statistical and mechanistic approaches (Lyu et al., 2023) detailed in Appendix G.6. We notably include a time-homogeneous ARPLN-ICA ((h)-ARPLN-ICA) as an ablation of regime modeling, a vector auto-regressive model VAR(1) suited to microbiome data (CLR-VAR(1), see Zheng & Chen (2017)), UwedgeICA (Pfister et al., 2019) and Picard (Ablin et al., 2018) as classical ICA preprocessing mixed with CLR-

VAR(1) for forecasting, an auto-regressive Poisson log-link model with two recurrent linear switching states (P-rSLDS, see Linderman et al. (2017)) and gLV-L2 (Stein et al., 2013) as a microbiome specialized mechanistic model. To ensure comparability across methods, all baselines are evaluated in the same offset setting, where predicted compositions are mapped back to count space using the observed test-time offset (see Appendix G.4). To assess forecasting performance, we evaluate complementary statistics that capture distributional accuracy for counts (Poisson deviance), pointwise prediction error (MAE log1p), and microbial community-level fidelity (Aitchison distance). Computations for each statistics are provided in Appendix H. Evaluation is performed through leave-one-out cross validation. Architecture and hyperparameters choices are discussed in Appendix G.5.

**Forecasting results.** Figure 6 summarizes forecasting performance across methods. ARPLN-ICA is competitive across metrics and achieves a clear gain in Poisson deviance, consistent with P-rSLDS which both match better the discrete count likelihood. On this task, explicitly modeling $C = 2$ regimes brings small gains, with the switching variant slightly outperforming its time-homogeneous counterpart on average. Overall, while PLN-ICA is not primarily designed for forecasting, these results indicate that it captures the longitudinal dynamics at a level comparable to forecasting-oriented models and remains competitive with ICA baselines, reinforcing its value as an interpretable analysis tool for temporal count data.

## 6. Illustration on climate data

### 6.1. Climate dataset and study design

**Storm Events dataset.** The NOAA Storm Events database records severe weather hazards occurring across the United States, yielding multivariate count profiles of significant or rare weather phenomena (NOAA et al., 1996). In this work, we consider yearly panels from 2000 to 2025

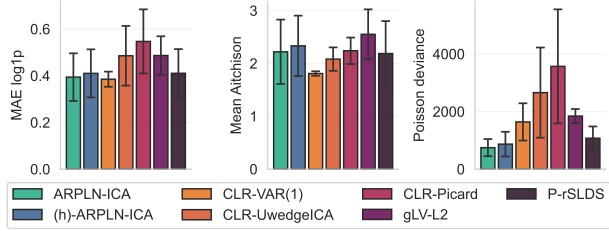

*Figure 6.* Average forecasting performance over history length $t_0 \in \{2, \ldots, 20\}$ on the gnotobiotic mice dataset. Boxplots show the distribution across LOO folds of the mean scoring value obtained by varying $t_0$ and forecasting the remainder of the trajectory. Lower is better for all metrics.

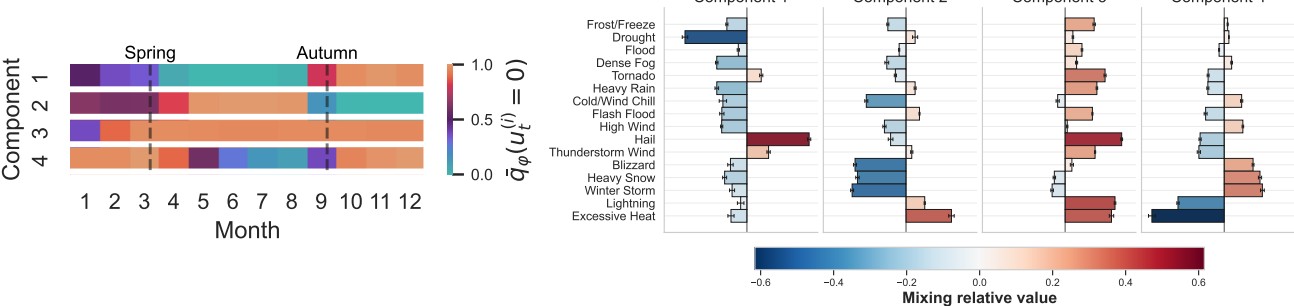

*Figure 7.* Regime switches and linear mixing loadings inferred by ARPLN-ICA on the Storm Events dataset. (Left) Average probability of being in latent state 0 for each component across LOO CV, see Appendix F.2. Components 1, 2, 4, highlight regime changes compatible with seasonal cycles. (Right) Weather event loading per component (column) of the medoid mixing across LOO CV, with standard deviation to the medoid across folds. Components highlight known seasonal patterns and meteorological contrasts at different scales.

$(n = 25)$ for $K = 16$ continental event types aggregated at the monthly scale $(T = 12)$. The resulting multivariate count time series exhibit pronounced heterogeneity across event types, with strong seasonal structure and many low or near-zero counts (see Appendix I.1). As dependencies between event types may reflect shared meteorological drivers, this dataset is well suited to analyzing recurrent co-occurrence patterns and regimes with ARPLN-ICA.

**Training procedure.** As in the microbiome experiments, our model is trained using LOO CV. We set $d = 4$ components as a trade-off between reconstruction quality and interpretability (see Appendix I.2). To mitigate climate change trends, ARPLN-ICA models are trained with plug-in log-sum offset estimators. To assess the transferability of our inference procedure, we use the same parameterization as in the microbiome experiments, namely $C = 2$ and the encoder architecture of Appendix 10, trained with AdamW as detailed in Appendix F.1. The stability of the ARPLN-ICA decomposition is further evaluated in Appendix I.2.

### 6.2. Severe weather events interactions and regimes

Figure 7 reports the ARPLN-ICA component-wise switching states across the year together with the corresponding mixing loadings. The inferred states are consistent with a broad warm–cold seasonal contrast, with components 1, 2, and 4 switching between spring and autumn, suggesting that the model captures large-scale seasonal structure. The ICA loadings further recover interpretable meteorological associations: component 1 contrasts severe convective hazards with moisture-related events (Barton et al., 2025); component 2 opposes excessive heat with winter-related hazards; component 3 captures a compound warm-season weather; and component 4 contrasts seasonally opposed synoptic winter hazards with heat-related convective events. Together, these results show that ARPLN-ICA can disentangle seasonal regime changes from weather hazard occurrences, providing interpretable representations of climate event.

## 7. Conclusion

In this work, we introduced a generative ICA framework for temporal count data, coupling count-specific modeling with perturbation-aware switching dynamics. To support interpretability, we established identifiability guarantees, complemented with finite-sample evidence in a linear ICA model setting. The practical value of the approach was illustrated on two temporal count datasets: a gut microbiome study, where the inferred regimes align with clinical perturbations and reveal biologically meaningful associations; and a severe weather hazard dataset, which recovered recurrent seasonal structure and known meteorological patterns. An auxiliary forecasting task on microbiome data further showed that ARPLN-ICA is competitive in predicting count trajectories, providing quantitative support for the qualitative analyses. Overall, the model yields principled representations of temporal count data, supporting its use for representation learning and exploratory analysis.

Meanwhile, our approach opens several research perspectives. Although Proposition 3.1 applies to nonlinear ICA, the present method focuses on linear mixing, where identifiability applies under mild assumptions and feature–source associations remain interpretable through the loadings. A natural extension is to consider richer nonlinear mixing, for instance by relaxing the assumptions of Theorem 2 in (Hälvä et al., 2024), while preserving forms of interpretability. Subsequently, parameter learning relies on amortized variational inference, which can induce an objective gap relative to maximum likelihood estimation (Shu et al., 2018). Establishing formal generalization guarantees for such variational procedures would therefore be valuable, in the spirit of Tang & Yang (2021). In addition, while our microbiome case study is encouraging, extending the framework to real-world cohorts remains nontrivial as clinical datasets often exhibit missing data which would require methodological developments tailored to these settings.

## Acknowledgements

The authors would like to thank Harry Sokol for his medical expertise on the microbiome analyses. The authors also thank the anonymous reviewers for their valuable feedback which have contributed to improve the manuscript.

## Impact Statement

This paper presents methodological contributions aimed at advancing machine learning research. We do not foresee specific societal risks beyond the potential for downstream misuse or misinterpretation of model outputs without appropriate domain expertise and validation. The ethical aspects of the gnotobiotic mice study were reviewed by the committees reported in Bucci et al. (2016), following institutional guidelines on animal experimentation protocols.

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

## Appendix organization and notations

The appendix is organized as follows:

- **Appendix A** presents the proofs of our identifiability results for PLN-ICA and ARPLN-ICA, and discusses their connection to prior work.

- **Appendix C** derives the optimal CAVI update for the switching labels, and provides background on structured variational inference and coordinate ascent variational inference.

- **Appendix D** derives the closed-form updates for the hyperparameters of the ARPLN-ICA prior.

- **Appendix E** provides additional details for the simulation study, including the mean-field approximation, baseline methods, experimental protocols, metrics definitions, and an analysis of scalability and wall-clock time.

- **Appendix F** describes the microbiome study design, the computation of empirical switching distributions, and discusses components stability and model selection, with a comparison to the main manuscript analysis at $d = 7$.

- **Appendix I** provides additional analyses on the Storm Events dataset, including model selection and stability.

**Notations.** In the following, $\odot$ denotes the pointwise product operator and $\oslash$ the pointwise division operator. The superscript $\cdot^*$ broadcasts a given vector in the appropriate dimension. The Kullback-Leibler divergence between two densities $q, p$ is denoted by $\text{KL}\,[q\|p]$. The entropy of a distribution $p$ is denoted by $\text{H}[p]$. When two random variables $x, y$ have the same distribution we note $x \sim y$.

## A. Identifiability results

Identifiability is a prerequisite for reliable inference and meaningful downstream interpretation (D'Amour et al., 2022). In this section, we establish the identifiability of PLN-ICA and ARPLN-ICA models under mild conditions. Proof of Proposition 3.1 is deferred to Appendix A.1, while proof of Corollary 3.2 is found in Appendix A.2.

### A.1. Proof of Proposition 3.1

The proof of Proposition 3.1 relies on two Lemmas. First, Lemma A.1 shows the identifiability of univariate PLN models when the latent variable has arbitrary temporal dependency. Lemma A.2 then extends this result to the multivariate setting.

**Lemma A.1.** *Let* $(x, z) = (x_t, z_t)_{1 \leq t \leq T}$ *and* $(\tilde{z}, \tilde{z}) = (\tilde{x}_t, \tilde{z}_t)_{1 \leq t \leq T}$ *such that* $z, \tilde{z} \in \mathbb{R}_{>0}^T$ *and* $(x_t)_{1 \leq t \leq T}$ *(resp.* $(\tilde{x}_t)_{1 \leq t \leq T}$*) are conditionally independent given* $(z_t)_{1 \leq t \leq T}$ *(resp.* $(\tilde{z}_t)_{1 \leq t \leq T}$*) and such that for all* $1 \leq t \leq T$, $x_t \sim \mathcal{P}(z_t)$ *(resp.* $\tilde{x}_t \sim \mathcal{P}(\tilde{z}_t)$*). Then, if* $x$ *and* $\tilde{x}$ *have the same distribution,* $z$ *and* $\tilde{z}$ *have the same distribution.*

**Lemma A.2.** *Let* $(x, z) = (x_t, z_t)_{1 \leq t \leq T}$ *(resp.* $(\tilde{x}, \tilde{z})$*) such that* $z \in \mathbb{R}_{>0}^{T \times K}$ *(resp.* $\tilde{z} \in \mathbb{R}_{>0}^{T \times K}$*) and* $(x_t^{(k)})_{1 \leq t \leq T, 1 \leq k \leq K}$ *are conditionally independent on their respective* $(z_t^{(k)})_{1 \leq t \leq T, 1 \leq k \leq K}$ *such that for all* $t \leq T, k \leq K, x_t^{(k)} \sim \mathcal{P}(z_t^{(k)})$ *(resp.* $(\tilde{x}_t^{(k)})_{1 \leq t \leq T}$ *with* $(\tilde{z}_t^{(k)})_{1 \leq t \leq T}$*). Then, if* $x$ *and* $\tilde{x}$ *have the same distribution,* $z$ *and* $\tilde{z}$ *have the same distribution.*

Then, defining $z = (\exp f(s_t))_{1 \leq t \leq T}$ and $\tilde{z} = (\exp \tilde{f}(\tilde{s}_t))_{1 \leq t \leq T}$, Lemma A.2 induces $z \sim \tilde{z}$. Taking the log point-wise yields $(f(s_t))_{1 \leq t \leq T} \sim (\tilde{f}(\tilde{s}_t))_{1 \leq t \leq T}$ concluding the proof of Proposition 3.1. $\square$

**Proof of Lemma A.1.** The proof follows the arguments of Lemma 5 of Chaussard et al. (2025a) with standard PLN models. Let $h$ a measurable function such that $h(x) = \prod_{t=1}^T h_t(x_t)$, then

$$\mathbb{E}[h(x)] = \mathbb{E}\left[\prod_{t=1}^T \sum_{u \in \mathbb{N}} e^{-z_t} \frac{z_t^u}{u!} h_t(u)\right] . \tag{8}$$

For all $1 \leq t \leq T$, choosing $\beta_t \in (-\infty, 1]$ and $h_t(u) = \beta_t^u$ yields

$$\mathbb{E}[h(x)] = \mathbb{E}\left[\prod_{t=1}^{T} e^{-z_t} \sum_{u \in \mathbb{N}} \frac{(z_t \beta_t)^u}{u!}\right] = \mathbb{E}\left[e^{-(1-\beta)^\top z}\right] = \mathcal{L}_z(1 - \beta), \tag{9}$$

where $\beta = (\beta_1, \ldots, \beta_T)$ and $\mathcal{L}_z$ denotes the Laplace transform of $z$. Thus, since $x$ and $\tilde{x}$ have the same distribution, the Laplace transforms of $z, \tilde{z}$ coincide on $\mathbb{R}_+^T$, that is for all $v \in [0, +\infty)^T$, $\mathcal{L}_z(v) = \mathcal{L}_{\tilde{z}}(v)$. Since $z, \tilde{z}$ are non-negative, their Laplace transform is finite on $\mathbb{R}_+^T$ and uniquely characterizes their distributions, which concludes the proof. $\qquad \square$

**Proof of Lemma A.2.** Consider the indexing $i(t, k) = (t-1)K + k$, then for the index $(t, k)$ we set $j = i(t, k)$ and define $x_j' = x_t^{(k)}$ and $z_j' = z_t^{(k)}$. By model definition, we get that $(x_j')_{1 \leq j \leq TK}$ are conditionally independent given $(z_j')_{1 \leq j \leq TK}$ with $x_j' \sim \mathcal{P}(z_j')$ conditionally on $z_j' \in \mathbb{R}_{>0}^{TK}$. Using a similar construction for $(\tilde{x}_j', \tilde{z}_j')_{1 \leq j \leq TK}$, since $x' \sim \tilde{x}'$ by Lemma A.1 we get $z' \sim \tilde{z}'$ hence $z \sim \tilde{z}$ concluding the proof. $\qquad \square$

## A.2. Proof of Corollary 3.2

From Proposition 3.1, we get $(\Gamma s_t)_{1 \leq t \leq T} \sim (\tilde{\Gamma}\tilde{s}_t)_{1 \leq t \leq T}$. Thus, the remaining indeterminacy lies in the identification of linear mixed ICA under regime switching, which is characterized in the following Lemma A.3.

**Lemma A.3.** Let $K \geq d > 0$ and $C \in \mathbb{N}^*$. Define $(w, s) = (w_t, s_t)_{1 \leq t \leq T}$ (resp. $(\tilde{w}, \tilde{s})$) such that $s \in \mathbb{R}^{T \times d}$ (resp. $\tilde{s}$) and $w \in \mathbb{R}^{T \times K}$ (resp. $\tilde{w}$), and let $\Gamma \in \mathbb{R}^{K \times d}$ (resp. $\tilde{\Gamma}$). Assume the model assumptions hold:

- $\Gamma$ (resp. $\tilde{\Gamma}$) satisfy $\text{rank}(\Gamma) = d$ and define $w_t = \Gamma s_t$ (resp. $\tilde{w}_t = \tilde{\Gamma}\tilde{s}_t$).

- The coordinate-wise collections $\left((u_t^{(i)})_{1 \leq t \leq T}, (s_t^{(i)})_{1 \leq t \leq T}\right)_{1 \leq i \leq d}$ (resp. $(\tilde{u}_t^{(i)})_{1 \leq t \leq T}, (\tilde{s}_t^{(i)})_{1 \leq t \leq T}$) are mutually independent across components $i$.

- For each $i \in \{1, \ldots, d\}$, let $(u_t^{(i)})_{1 \leq t \leq T}$ (resp. $(\tilde{u}_t^{(i)})_{1 \leq t \leq T}$) be a discrete Markov chain on $\{1, \ldots, C\}$ with initial law $\pi^{(i)}$ and transition matrix $A^{(i)}$ (resp. $\tilde{\pi}^{(i)}, \tilde{A}^{(i)}$).

- Assume that, conditionally on $\{u_1^{(i)} = k\}$, $s_1^{(i)} \sim \mathcal{N}(m_k^{(i)}, \Phi_k^{(i)})$, and conditionally on $\{s_t^{(i)}, u_{t+1}^{(i)} = k\}$, $s_{t+1}^{(i)} \sim \mathcal{N}(B_k^{(i)} s_t^{(i)} + b_k^{(i)}, \Psi_k^{(i)})$, where for all $k$, $B_k \in D_d(\mathbb{R}^*)$ and $\Phi_k, \Psi_k \in D_d(\mathbb{R}_{>0})$. Assume a similar construction for $\tilde{s}$ with parameters $\tilde{\pi}^{(i)}, \tilde{A}^{(i)}, \tilde{m}_k^{(i)}, \tilde{\Phi}_k^{(i)}, \tilde{B}_k^{(i)}, \tilde{b}_k^{(i)}, \tilde{\Psi}_k^{(i)}$.

Then, assuming that $w$ and $\tilde{w}$ have the same distribution and that $\Sigma_{1,1} = \mathbb{C}\text{ov}(s_1, s_1)$ has positive diagonal entries, if there exist $t_0 \in \{1, \ldots, T\}$ and $\ell_0 \in \{0, \ldots, t_0 - 1\}$ such that the diagonal entries of

$$\Sigma_{1,1}^{-1/2} \Sigma_{t_0, t_0 - \ell_0} \Sigma_{1,1}^{-1/2} \quad \text{where } \Sigma_{t,s} = \mathbb{C}\text{ov}(s_t, s_s) \tag{10}$$

are pairwise distinct, then there exist a permutation matrix $P \in \mathbb{R}^{d \times d}$ and a diagonal scaling $D \in D_d(\mathbb{R}^*)$ such that

$$\tilde{\Gamma} = \Gamma P D^{-1}. \tag{11}$$

*Proof.* We first show the result with $C = 1$ under milder assumptions in Appendix A.3. The proof with $C > 1$ relies on identification arguments from Belouchrani et al. (2002), and is postponed to Appendix A.4. $\qquad \square$

## A.3. Identifiability of regime-homogeneous ARPLN-ICA

We consider the regime-homogeneous ARPLN-ICA characterized by $C = 1$ (no switching). In this setting, Corollary A.4 shows that the mixing function can be identified under mild conditions on the AR coefficients of the underlying dynamic. The proof relies on Proposition 3.1 combined with a similar proof strategy as that of Theorem 1 of Belouchrani et al. (2002) reported in Lemma A.5, specialized to Gaussian AR(1) models.

**Corollary A.4.** Let $(x, s) = (x_t, s_t)_{1 \leq t \leq T}$ (resp. $(\tilde{x}, \tilde{s})$) such that $x \in \mathbb{N}^{T \times K}$, $s \in \mathbb{R}^{T \times d}$ (resp. $\tilde{x}, \tilde{s}$) with $K \geq d > 0$. Assume $(x, s)$, $(\tilde{x}, \tilde{s})$ are distributed according to two distinct ARPLN-ICA models with $C = 1$. Assume $\Gamma \in \mathbb{R}^{K \times d}$ (resp. $\tilde{\Gamma}$) is full rank and $B$ (resp. $\tilde{B}$) is diagonal with non-zero pairwise distinct entries. Then, if $x$ and $\tilde{x}$ have the same distribution, $\Gamma$ and $\tilde{\Gamma}$ are equal up to column-wise permutation and non-zero rescaling.

*Proof.* From Proposition 3.1, we get $(\Gamma s_t)_{1 \le t \le T} \sim (\tilde{\Gamma} \tilde{s}_t)_{1 \le t \le T}$. The remaining indeterminacies are then reduced to identification of linear ICA models, characterized in the following Lemma:

**Lemma A.5.** *Let* $(w, s) = (w_t, s_t)_{1 \le t \le T}$ *(resp.* $(\tilde{w}, \tilde{s})$*) such that* $s \in \mathbb{R}^{T \times d}$ *(resp.* $\tilde{s} \in \mathbb{R}^{T \times d}$*) and* $w \in \mathbb{R}^{T \times K}$ *(resp.* $\tilde{w} \in \mathbb{R}^{T \times K}$*) with* $K \ge d > 0$*. Assume that* $s$ *is a d-dimensional Markov chain such that* $s_1 \sim \mathcal{N}(m, \Phi)$ *and, conditionally on* $s_t$*,* $s_{t+1} \sim \mathcal{N}(Bs_t + b, \Psi)$*, where* $B \in D_d(\mathbb{R}^*)$ *has pairwise distinct diagonal entries, and* $\Phi, \Psi \in D_d(\mathbb{R}_{>0})$*. Define the observations at* $t \le T$ *by* $w_t = \Gamma s_t$*, where* $\Gamma \in \mathbb{R}^{K \times d}$ *satisfies* $\text{rank}(\Gamma) = d$*. Assume a similar construction for* $(\tilde{w}, \tilde{s})$ *with Markov parameters* $\tilde{B}, \tilde{\Phi}, \tilde{\Psi} \in D_d(\mathbb{R}^*)$ *and* $\tilde{\Gamma}$ *of rank d. Then, if* $w$ *and* $\tilde{w}$ *have the same distribution, there exists a permutation matrix* $P \in \mathbb{R}^{d \times d}$ *and a diagonal scaling matrix* $D \in D_d(\mathbb{R}^*)$ *such that*

$$\tilde{\Gamma} = \Gamma P D^{-1} , \quad \tilde{B} = D P^\top B P D^{-1} , \quad \tilde{\Phi} = D P^\top \Phi P D ,$$
$$\tilde{\Psi} = D P^\top \Psi P D , \quad \tilde{b} = D P^\top b , \quad \tilde{m} = D P^\top m . \tag{12}$$

concluding the proof. $\square$

**Proof of Lemma A.5.** Linear mixing identification has been widely studied in ICA and blind source identification. In classical linear ICA models, identifiability of the mixing matrix relies on additional structure such as non-Gaussianity. In particular, when the latent sources are Gaussian and no model structure is exploited, the mixing matrix is not identifiable beyond an arbitrary orthogonal transformation (Hyvärinen & Oja, 2000). When temporal dependence is available, second-order methods such as AMUSE (Tong et al., 1990) and SOBI (Belouchrani et al., 2002) leverage temporal auto-correlation structure to identify the mixing matrix up to permutation and column-wise rescaling. Their generic identifiability conditions typically require sufficiently distinct second-order statistics across components, either at a single-lag (see their Theorem 1) or across joint diagonalization (see their Theorem 2). For completeness, we provide a self-contained derivation showing that these identifiability arguments specialize to the latent linear-Gaussian dynamics in ARPLN-ICA, and yield the same invariance class in our setting, with weaker assumptions in the regime-homogeneous AR models given in Lemma A.5.

*Proof.* Let $v$ (resp. $\tilde{v}$) the centered process of $s_t$ defined by

$$v_t = s_t - \mathbb{E}[s_t] . \tag{13}$$

Denote the auto-correlation matrix at time $t, \ell \ge 0$ by

$$\Sigma_{t, t-\ell} = \mathbb{C}\text{ov}(v_t, v_{t-\ell}) . \tag{14}$$

Since $s_t$ is component-wise independent, $v_t$ is component wise independent and thus $\Sigma_{t, t-\ell}$ is diagonal for all $t, \ell \ge 0$. Additionally, the assumption $w_t \sim \tilde{w}_t$ yields $\Gamma s_t \sim \tilde{\Gamma} \tilde{s}_t$, and thus $\Gamma v_t \sim \tilde{\Gamma} \tilde{v}_t$, which yields

$$\Gamma \Sigma_{t, t-\ell} \Gamma^\top = \tilde{\Gamma} \tilde{\Sigma}_{t, t-\ell} \tilde{\Gamma}^\top . \tag{15}$$

In particular, for $t = 1$, $\ell = 0$ we get $\Gamma \Phi \Gamma^\top = \tilde{\Gamma} \tilde{\Phi} \tilde{\Gamma}^\top$ where $\Phi, \tilde{\Phi}$ are diagonal positive definite covariance matrices, and thus of rank $d$. Consequently, by Lemma A.6, there exists $Q \in \mathbb{R}^{d \times d}$ invertible such that

$$\tilde{\Gamma} = \Gamma Q . \tag{16}$$

Since $\Gamma$ is full rank, there exists a left inverse $\Gamma^\dagger \in \mathbb{R}^{d \times K}$ such that $\Gamma^\dagger \Gamma = I_d$. Applying the left inverse to Eq. (15) and using Eq. (16) yields

$$\Sigma_{t, t-\ell} = Q \tilde{\Sigma}_{t, t-\ell} Q^\top . \tag{17}$$

Let $L_1, \tilde{L}_1$ the Cholesky decomposition of $\Sigma_{1,1} = \Phi, \tilde{\Sigma}_{1,1} = \tilde{\Phi}$ respectively. Since both are diagonal positive definite matrices, the Cholesky decomposition are diagonal positive definite, and thus we can define $F = L_1^{-1} Q \tilde{L}_1$. Using that $\Sigma_{1,1} = L_1 L_1^\top$ and $\tilde{\Sigma}_{1,1} = \tilde{L}_1 \tilde{L}_1^\top$ in Eq. (17) yields $F$ orthogonal since

$$I_d = (L_1^{-1} Q \tilde{L}_1)(L_1^{-1} Q \tilde{L}_1)^\top = F F^\top . \tag{18}$$

Focusing on the auto-correlation at $t = 2$, $\ell = 1$, using the tower property we have

$$\Sigma_{2,1} = \mathbb{C}\text{ov}(v_2, v_1) = \mathbb{E}[v_2 v_1^\top] = \mathbb{E}[\mathbb{E}[v_2 \mid v_1] v_1^\top] = B\Phi = B L_1 L_1^\top . \tag{19}$$

Thus using this decomposition in Eq. 17 gives $BL_1L_1 = Q\tilde{B}\tilde{L}_1\tilde{L}_1Q^\top$. Since $B, L_1, \tilde{B}, \tilde{L}_1$ are diagonal, they can be permuted in preferred order, and thus by invertibility of $L_1$ we get

$$B = (L_1^{-1}Q\tilde{L}_1)\tilde{B}(L_1^{-1}Q\tilde{L}_1)^\top = F\tilde{B}F^\top . \tag{20}$$

Since $B, \tilde{B}$ are diagonal with non-zero entries and $F$ orthogonal, by Lemma A.7 $F$ is the composition of a permutation $P \in \mathbb{R}^{d\times d}$ with a signed diagonal matrix $S = [\pm e_i]_{1\leq i\leq d}$ where $(e_i)_{1\leq i\leq d}$ denotes the canonical basis, and thus $F = PS$. Additionally, since $F = L_1^{-1}Q\tilde{L}_1$ we get that $Q = PD^{-1}$ with $D^{-1} = (P^\top L_1 P)S\tilde{L}_1^{-1}$ diagonal non-zero since $L_1, \tilde{L}_1, S$ are diagonal with non-zero entries. Consequently, since $\tilde{\Gamma} = \Gamma Q$ we get

$$\tilde{\Gamma} = \Gamma PD^{-1} . \tag{21}$$

Then, using that $w_1 \sim \tilde{w}_1$, $\Gamma^\dagger\Gamma = I_d$, and $Q^{-1} = (PD^{-1})^{-1} = DP^\top$, we have

$$\tilde{m} = DP^\top m , \quad \tilde{\Phi} = DP^\top \Phi PD . \tag{22}$$

Similarly, using that conditionally on $w_1, \tilde{w}_1, \Gamma v_2 \sim \tilde{\Gamma}\tilde{v}_2$, we have

$$\tilde{\Psi} = DP^\top \Psi PD . \tag{23}$$

Then, using Eq. (19) and Eq. (17) we get $BQ = Q\tilde{B}$ and thus

$$\tilde{B} = DP^\top BPD^{-1} . \tag{24}$$

Finally, the bias terms $b, \tilde{b}$ are recovered from the processes $w_2, \tilde{w}_2$ which have the same distribution conditionally on $w_1, \tilde{w}_1$, and thus same mean that is

$$Bm + b = PD^{-1}\tilde{B}\tilde{m} + PD^{-1}\tilde{b} . \tag{25}$$

Since $B = PD^{-1}\tilde{B}DP^\top$, using that $\tilde{m} = DP^\top m$ we get $PD^{-1}\tilde{B}\tilde{m} = PD^{-1}\tilde{B}DP^\top m = Bm$, which yields

$$\tilde{b} = DP^\top b , \tag{26}$$

concluding the proof. $\qquad\square$

## A.4. Proof of Lemma A.3: Identifiability of linear switching dynamic with linear mixing

We extend the result from the regime-homogeneous setting of Lemma A.5 to the linear switching dynamics setting. In the regime-homogeneous case, identification can be achieved by exploiting the AR coefficient matrix $B$, but this argument does not apply when considering regime changes. We therefore adopt the classical second-order whitening approach of Belouchrani et al. (2002) to establish identifiability of the linear mixing component in ARPLN-ICA. Importantly, the joint diagonalization assumption on lagged covariances required in Belouchrani et al. (2002) can be relaxed in our setting thanks to the sources independence, yielding milder conditions than those required in their setting.

*Proof.* The proof follows the second-order argument of Lemma A.5. Let $v$ define the centered process $v_t = s_t - \mathbb{E}[s_t]$. Since $w \sim \tilde{w}$, we have $\Gamma v_t \sim \tilde{\Gamma}\tilde{v}_t$. For all $t, \ell$, consider the auto-correlation

$$\Sigma_{t,t-\ell} = \mathbb{C}\mathrm{ov}(v_t, v_{t-\ell}), \tag{27}$$

and similarly $\tilde{\Sigma}_{t,t-\ell} = \mathbb{C}\mathrm{ov}(\tilde{v}_t, \tilde{v}_{t-\ell})$. Then for all $t, \ell \geq 0$, since $\Gamma v_t \sim \tilde{\Gamma}\tilde{v}_t$ we have

$$\Gamma\Sigma_{t,s}\Gamma^\top = \tilde{\Gamma}\tilde{\Sigma}_{t,s}\tilde{\Gamma}^\top . \tag{28}$$

By definition of the linear switching dynamic model, the latent process factorizes as $p_{\boldsymbol{\theta}}(s, u) = \prod_{i=1}^d p_{\boldsymbol{\theta}}(s_{1:T}^{(i)}, u_{1:T}^{(i)})$. Therefore, marginalizing on $u$ yields $p_{\boldsymbol{\theta}}(s) = \prod_{i=1}^d p_{\boldsymbol{\theta}}(s_{1:T}^{(i)})$, indicating that components are mutually independent and thus, for every $t, \ell, \Sigma_{t,t-\ell}$ is diagonal. In particular $\Sigma_{1,1}$ is diagonal with strictly positive entries by assumption and therefore has a diagonal Cholesky factor $L_1 \in D_d(\mathbb{R}_+^*)$ such that $\Sigma_{1,1} = L_1L_1^\top$, and similarly $\tilde{\Sigma}_{1,1} = \tilde{L}_1\tilde{L}_1^\top$ with $\tilde{L}_1 \in D_d(\mathbb{R}_+^*)$.

Taking $t = 1, \ell = 0$ in (28) yields

$$\Gamma \Sigma_{1,1} \Gamma^\top = \tilde{\Gamma} \tilde{\Sigma}_{1,1} \tilde{\Gamma}^\top .$$

Since $\Sigma_{1,1}$ and $\tilde{\Sigma}_{1,1}$ are diagonal positive definite covariance matrices, they are full rank. Then, Lemma A.6 implies that there exists an invertible matrix $Q \in \mathbb{R}^{d \times d}$ such that

$$\tilde{\Gamma} = \Gamma Q . \tag{29}$$

Letting $\Gamma^\dagger$ a left-inverse of $\Gamma$ then yields for all $t, \ell \geq 0$,

$$\Sigma_{t,t-\ell} = Q \tilde{\Sigma}_{t,t-\ell} Q^\top . \tag{30}$$

Define $L_1, \tilde{L}_1$ the respective diagonal positive definite Cholesky of the covariances $\Sigma_{1,1}, \tilde{\Sigma}_{1,1}$, then by the same arguments as in Proposition A.5, $F = L_1^{-1} Q \tilde{L}_1$ is orthogonal. Let $(t_0, \ell_0)$ satisfy (10), we define $G = L_1^{-1} \Sigma_{t_0,t_0-\ell_0} L_1^{-\top}$ (resp. $\tilde{G}$), diagonal by construction. Since by Eq. (30), we obtain

$$G = L_1^{-1} Q \tilde{\Sigma}_{t_0,t_0-\ell_0} Q^\top L_1^{-\top} = (L_1^{-1} Q \tilde{L}_1)(\tilde{L}_1^{-1} \tilde{\Sigma}_{t_0,t_0-\ell_0} \tilde{L}_1^{-\top})(L_1^{-1} Q \tilde{L}_1)^\top = F \tilde{G} F^\top . \tag{31}$$

By assumption, $G$ has pairwise distinct diagonal entries and $F$ is orthogonal. Therefore, by Lemma A.7, there exist a permutation matrix $P \in \mathbb{R}^{d \times d}$ and a signed diagonal matrix $S \in D_d(\{-1, 1\})$ such that $F = PS$. The proof is concluded by the same arguments as that of Lemma A.5. □

## A.5. Technical lemmas

**Lemma A.6.** *Let $\Sigma, \tilde{\Sigma} \in D_d(\mathbb{R}_+^*)$ positive diagonal invertible and $\Gamma, \tilde{\Gamma} \in \mathbb{R}^{K \times d}$ of rank d. If $\Gamma \Sigma \Gamma^\top = \tilde{\Gamma} \tilde{\Sigma} \tilde{\Gamma}^\top$, then there exists $Q \in \mathbb{R}^{d \times d}$ invertible such that $\tilde{\Gamma} = \Gamma Q$.*

*Proof.* Let $R = \Gamma \Sigma \Gamma^\top = \tilde{\Gamma} \tilde{\Sigma} \tilde{\Gamma}^\top$, then by definition $\mathrm{Im}(R) \subseteq \mathrm{Im}(\Gamma)$ and $\mathrm{Im}(R) \subseteq \mathrm{Im}(\tilde{\Gamma})$.

Since $\Sigma$ is positive diagonal, we define its Cholesky $L = \Sigma^{1/2} = L^\top$. Then, since $\Gamma L$ is a real-valued matrix $\mathrm{rank}(R) = \mathrm{rank}(\Gamma \Sigma \Gamma^\top) = \mathrm{rank}(\Gamma L)$. Additionally, $L$ is of rank $d$ since $\Sigma$ is of rank $d$, thus $\mathrm{rank}(R) = \mathrm{rank}(\Gamma) = d$ which yields $\mathrm{Im}(R) = \mathrm{Im}(\Gamma) = \mathrm{Im}(\tilde{\Gamma})$.

Thus $\Gamma, \tilde{\Gamma}$ have the same column space, that is for some $Q \in \mathbb{R}^{d \times d}, \tilde{\Gamma} = \Gamma Q$. Thus, $\mathrm{rank}(\tilde{\Gamma}) = \mathrm{rank}(\Gamma Q) \leq \min\{\mathrm{rank}(\Gamma), \mathrm{rank}(Q)\} \leq \mathrm{rank}(Q)$, and since $\mathrm{rank}(\Gamma) = \mathrm{rank}(\tilde{\Gamma}) = d$ this forces $\mathrm{rank}(Q) = d$, concluding the proof. □

**Lemma A.7.** *Let $F \in \mathbb{R}^{d \times d}$ an orthogonal matrix, $A, B \in D_d(\mathbb{R}^*)$ two diagonal matrices with non-zero diagonal coefficients. Then, if $A = FBF^\top$, there exists a permutation matrix $P \in \mathbb{R}^{d \times d}$ and a signed identity matrix $S = \mathrm{diag}([\pm 1, \ldots, \pm 1]) \in \mathbb{R}^{d \times d}$ such that $F = PS$.*

*Proof.* Since $F^{-1} = F^\top$ and $A = FBF^\top$, $F$ is a change-of-basis matrix. Thus if $\{a_1, \ldots, a_d\}$ denotes the specter of $A$ and $\{b_1, \ldots, b_d\}$ the specter of $B$ we have $\{a_1, \ldots, a_d\} = \{b_1, \ldots, b_d\}$.

Besides, $A = FBF^\top$ yields $AF = FB$ and thus denoting by $[f_j]_{1 \leq j \leq d}$ the columns of $F$, for all $j \leq d$

$$A f_j = b_j f_j , \tag{32}$$

that is $(f_j)_{1 \leq j \leq d}$ are eigen-vectors of $A$ with eigen-values $(b_j)_{1 \leq j \leq d}$. Since $A$ is diagonal, denoting by $(e_k)_{1 \leq k \leq d}$ the canonical basis of $\mathbb{R}^d$ we get for all $k \leq d$,

$$A e_k = a_k e_k . \tag{33}$$

Thus for $j \leq d$, there exists $k \leq d$ such that

$$a_k e_k = b_j f_j . \tag{34}$$

Since $a_k, b_j$ are non-zero, we can denote $\alpha_{kj} = a_k / b_j \neq 0$, and thus for all $f_j = \alpha_{kj} e_k$. Since $F$ is orthogonal, for all $j \leq d, \|f_j\|_2^2 = 1$ and thus $\alpha_{kj} = \pm 1$, concluding the proof. □

# B. Evidence Lower Bound derivation

Given temporal count data $x = x_{1:T}^{1:K} \in \mathbb{N}^{T \times K}$, the ARPLN-ICA model induces an intractable observed likelihood $p_{\boldsymbol{\theta}}(x)$, preventing regular maximum likelihood estimation. Similarly, the posterior $p_{\boldsymbol{\theta}}(s, u \mid x)$ is intractable, preventing the use of the Expectation-Maximization estimation algorithm. Consequently, we perform parameter inference using a variational proxy (Section 3.3). For the variational family specified in Eq. (5) and Eq. (6), parameter learning amounts to maximizing the evidence lower bound (ELBO) over $(\boldsymbol{\theta}, \boldsymbol{\varphi})$ given by

$$\mathcal{L}(x; \boldsymbol{\theta}, \boldsymbol{\varphi}) = \mathbb{E}_{q_{\boldsymbol{\varphi}}(\cdot|x)}[\log p_{\boldsymbol{\theta}}(x, s, u) - \log q_{\boldsymbol{\varphi}}(s, u \mid x)] \,. \tag{35}$$

Alternatively, the ELBO can be expressed as

$$\mathcal{L}(x; \boldsymbol{\theta}, \boldsymbol{\varphi}) = \mathbb{E}_{q_{\boldsymbol{\varphi}}(\cdot|x)}[\log p_{\boldsymbol{\theta}}(x \mid s, u)] - \mathbb{E}_{q_{\boldsymbol{\varphi}}(\cdot|x)}\left[\log \frac{q_{\boldsymbol{\varphi}}(s, u \mid x)}{p_{\boldsymbol{\theta}}(s, u)}\right] \,, \tag{36}$$

which then yields the particular decomposition for ARPLN-ICA models given by

$$\mathcal{L}(x; \boldsymbol{\theta}, \boldsymbol{\varphi}) = \mathbb{E}_{q_{\boldsymbol{\varphi}}(\cdot|x)}\left[\log p_{\boldsymbol{\theta}}(x_{1:T}^{(1:K)} \mid s_{1:T}^{(1:d)})\right] - \mathbb{E}_{q_{\boldsymbol{\varphi}}(\cdot|x)}\left[\log \frac{q_{\boldsymbol{\varphi}}(s_{1:T}^{(1:d)}, u_{1:T}^{(1:d)} \mid x)}{p_{\boldsymbol{\theta}}(s_{1:T}^{(1:d)} \mid u_{1:T}^{(1:d)})p_{\boldsymbol{\theta}}(u_{1:T}^{(1:d)})}\right]$$

$$= \sum_{t=1}^{T} \mathbb{E}_{q_{\boldsymbol{\varphi}}(\cdot|x)}[\log p_{\boldsymbol{\theta}}(x_t \mid s_t)] - \sum_{i=1}^{d} \left[\mathrm{KL}\left[q_{\boldsymbol{\varphi}}(u_{1:T}^{(i)} \mid x)\|p_{\boldsymbol{\theta}}(u_{1:T}^{(i)})\right] - \mathrm{H}\left[q_{\boldsymbol{\varphi}}(s_{1:T}^{(i)} \mid x)\right]\right. \tag{37}$$

$$\left. - \mathbb{E}_{q_{\boldsymbol{\varphi}}(\cdot|x)}\left[\log p_{\boldsymbol{\theta}}(s_1^{(i)} \mid u_1^{(i)})\right] - \sum_{t=1}^{T-1} \mathbb{E}_{q_{\boldsymbol{\varphi}}(\cdot|x)}\left[\log p_{\boldsymbol{\theta}}(s_{t+1}^{(i)} \mid u_{t+1}^{(i)}, s_t^{(i)})\right]\right] \,.$$

Each term can be derived in closed-form, starting with the variational PLN emission term given by

$$\mathbb{E}_{q_{\boldsymbol{\varphi}}(\cdot|x)}[\log p_{\boldsymbol{\theta}}(x_t \mid s_t)] = \sum_{i=1}^{K} \mathbb{E}_{q_{\boldsymbol{\varphi}}(\cdot|x)}\left[-\exp(\Gamma_i^\top s_t) + x_t^{(i)}\Gamma_i^\top s_t - \log x_t^{(i)}!\right]$$

$$= \sum_{i=1}^{K} x_t^{(i)}\Gamma_i^\top \mu_t(x) - \exp\left(\Gamma_i^\top \mu_t(x) + \frac{1}{2}\Gamma_i^\top \mathrm{diag}(\Sigma_t(x))\Gamma_i\right) - \log(x_t^{(i)}!) \,, \tag{38}$$

$$= 1_K^\top \left(x_t \odot (\Gamma\mu_t(x)) - \exp\left(\Gamma\mu_t(x) + \frac{1}{2}\mathrm{diag}(\Gamma\Sigma_t(x)\Gamma^\top)\right) - \log(x_t!)\right) \,,$$

where $\mu_t(x) = \mathbb{E}_{q_{\boldsymbol{\varphi}}(\cdot|x)}[s_t]$ and $\Sigma_t(x) = \mathbb{E}_{q_{\boldsymbol{\varphi}}(\cdot|x)}\left[(s_t - \mu_t)(s_t - \mu_t)^\top\right]$ are given by tower property recursions from the variational sources law, such that

$$\begin{aligned}
\mu_1(x) &= m(x), \quad \mu_t(x) = \tilde{B}_t(x)\mu_{t-1}(x) + \tilde{b}_t(x) \\
\Sigma_1(x) &= \tilde{\psi}_1(x), \quad \Sigma_t(x) = \tilde{\psi}_t(x) + \tilde{B}_t(x)^2\Sigma_{t-1}(x) \,,
\end{aligned} \tag{39}$$

where $m(\cdot), \tilde{B}_t(\cdot), \tilde{b}_t(\cdot), \tilde{\psi}_t(\cdot)$ are the variational parameters in Eq. (6), and where $m(\cdot), \tilde{b}_t(\cdot) \in \mathbb{R}^d$, $\tilde{B}_t(\cdot) \in \mathbb{R}^{d \times d}$ are diagonal and $\tilde{\psi}_t(\cdot)$ is diagonal positive definite. Then, denoting the variational switching states Markov kernels by

$$\nu_k^{(i)}(x) = q_{\boldsymbol{\varphi}}(u_1^{(i)} = k \mid x) \,, \quad \tau_{t+1,k\ell}^{(i)}(x) = q_{\boldsymbol{\varphi}}(u_{t+1}^{(i)} = k \mid u_t^{(i)} = \ell, x) \,, \tag{40}$$

the marginal variational distribution of switching states can be recursively expressed as

$$\begin{aligned}
\alpha_{1,k}^{(i)}(x) &= q_{\boldsymbol{\varphi}}(u_1^{(i)} = k \mid x) = \nu_k^{(i)}(x) \,, \\
\alpha_{t,k}^{(i)}(x) &= q_{\boldsymbol{\varphi}}(u_t^{(i)} = k \mid x) = \sum_{\ell=1}^{C} \tau_{t,k\ell}^{(i)}\alpha_{t-1,\ell}^{(i)}(x) \,.
\end{aligned} \tag{41}$$

Combining Eq. (40)-(41), the Kullback-Leibler divergence between the discrete switching states distributions is given by

$$
\begin{aligned}
\mathrm{KL}\left[q_{\boldsymbol{\varphi}}(u_{1:T}^{(i)} \mid x) \| p_{\boldsymbol{\theta}}(u_{1:T}^{(i)})\right] &= \sum_{u^{(i)} \in \{1,\ldots,C\}^T} q_{\boldsymbol{\varphi}}(u^{(i)} \mid x) \log \frac{q_{\boldsymbol{\varphi}}(u^{(i)} \mid x)}{p_{\boldsymbol{\theta}}(u^{(i)})} \\
&= \sum_{u^{(i)} \in \{1,\ldots,C\}^T} \Bigg[ q_{\boldsymbol{\varphi}}(u_1^{(i)} \mid x) \log \frac{q_{\boldsymbol{\varphi}}(u_1^{(i)} \mid x)}{p(u_1^{(i)})} \\
&\qquad\qquad + \sum_{t=1}^{T-1} q_{\boldsymbol{\varphi}}(u_{t+1}^{(i)} \mid u_t^{(i)}, x) q_{\boldsymbol{\varphi}}(u_t^{(i)} \mid x) \log \frac{q_{\boldsymbol{\varphi}}(u_{t+1}^{(i)} \mid u_t^{(i)}, x)}{p_{\boldsymbol{\theta}}(u_{t+1}^{(i)} \mid u_t^{(i)})} \Bigg] \\
&= \sum_{k=1}^{C} \nu_k^{(i)}(x) \log \frac{\nu_k^{(i)}(x)}{\pi_k^{(i)}} + \sum_{t=1}^{T-1} \sum_{k=1}^{C} \sum_{\ell=1}^{C} \tau_{t+1,k\ell}^{(i)}(x) \alpha_{t,\ell}^{(i)}(x) \log \frac{\tau_{t+1,k\ell}^{(i)}(x)}{a_{k\ell}^{(i)}} \, .
\end{aligned}
\tag{42}
$$

Then, the entropy of the variational distribution of the sources $q_{\boldsymbol{\varphi}}(s_{1:T}^{(i)} \mid x)$ is explicable as

$$
\begin{aligned}
\mathrm{H}\left[q_{\boldsymbol{\varphi}}(s_{1:T}^{(i)} \mid x)\right] &= \mathrm{H}\left[q_{\boldsymbol{\varphi}}(s_1^{(i)} \mid x)\right] + \sum_{t=1}^{T-1} \mathbb{E}_{q_{\boldsymbol{\varphi}}(\cdot|x)}\left[\mathrm{H}\left[q_{\boldsymbol{\varphi}}(s_{t+1}^{(i)} | s_t^{(i)}, x)\right]\right] \\
&= \frac{T}{2} \log 2\pi e + \frac{1}{2}\left(\log \tilde{\psi}_1^{(i)}(x) + \sum_{t=1}^{T-1} \log \tilde{\psi}_{t+1}^{(i)}(x)\right) ,
\end{aligned}
\tag{43}
$$

and thus, summing over all components yields

$$
\sum_{i=1}^{d} \mathrm{H}\left[q_{\boldsymbol{\varphi}}(s_{1:T}^{(i)} \mid x)\right] = \frac{Td}{2} \log 2\pi e + \frac{1}{2} \sum_{t=1}^{T} \log \det \tilde{\psi}_t(x) \, .
\tag{44}
$$

Finally, the expectation of the latent conditional likelihood can be written as

$$
\begin{aligned}
\mathbb{E}_{q_{\boldsymbol{\varphi}}(\cdot|x)}\left[\log p_{\boldsymbol{\theta}}(s_{t+1}^{(i)} | u_{t+1}^{(i)}, s_t^{(i)})\right] &= -\frac{1}{2}\mathbb{E}_{q_{\boldsymbol{\varphi}}(\cdot|x)}\left[\log 2\pi \psi_{u_{t+1}^{(i)}}^{(i)} + (\psi_{u_{t+1}^{(i)}}^{(i)})^{-1}\left(s_{t+1}^{(i)} - B_{u_{t+1}^{(i)}}^{(i)} s_t^{(i)} - b_{u_{t+1}^{(i)}}^{(i)}\right)^2\right] \\
&= -\frac{1}{2}\sum_{k=1}^{C} \alpha_{t+1,k}^{(i)}(x)\left(\log 2\pi \psi_k^{(i)} + (\psi_k^{(i)})^{-1}\mathbb{E}_{q_{\boldsymbol{\varphi}}(\cdot|x)}\left[\left(s_{t+1}^{(i)} - B_k^{(i)} s_t^{(i)} - b_k^{(i)}\right)^2\right]\right) \\
&= -\frac{1}{2}\sum_{k=1}^{C} \alpha_{t+1,k}^{(i)}(x)\bigg(\log 2\pi \psi_k^{(i)} + (\psi_k^{(i)})^{-1}\Big((\mu_{t+1}^{(i)}(x) - B_k^{(i)} \mu_t^{(i)}(x) - b_k^{(i)})^2 \\
&\qquad\qquad + \Sigma_{t+1}^{(i,i)}(x) + (B_k^{(i)})^2 \Sigma_t^{(i,i)}(x) - 2B_k^{(i)} \Sigma_{t,t+1}^{(i,i)}(x)\Big)\bigg) ,
\end{aligned}
\tag{45}
$$

where the lag-one auto-correlation term is computed using the tower property and mean recursion of Eq. (39) as

$$
\begin{aligned}
\Sigma_{t,t+1}^{(i,i)} &= \mathbb{E}_{q_{\boldsymbol{\varphi}}(\cdot|x)}\left[\left(s_{t+1}^{(i)} - \mu_{t+1}^{(i)}(x)\right)\left(s_t^{(i)} - \mu_t^{(i)}(x)\right)\right] \\
&= \mathbb{E}_{q_{\boldsymbol{\varphi}}(\cdot|x)}\left[\mathbb{E}_{q_{\boldsymbol{\varphi}}(\cdot|x)}\left[s_{t+1}^{(i)} - \mu_{t+1}^{(i)}(x) \mid s_t^{(i)}\right]\left(s_t^{(i)} - \mu_t^{(i)}(x)\right)\right] \\
&= \mathbb{E}_{q_{\boldsymbol{\varphi}}(\cdot|x)}\left[\tilde{B}_{t+1}^{(i)}(x)\left(s_t^{(i)} - \mu_t^{(i)}(x)\right)\left(s_t^{(i)} - \mu_t^{(i)}(x)\right)\right] \\
&= \tilde{B}_{t+1}^{(i)}(x)\Sigma_t^{(i,i)}(x) \, .
\end{aligned}
\tag{46}
$$

Similarly, the initial expectation can be computed as

$$
\begin{aligned}
\mathbb{E}_{q_{\boldsymbol{\varphi}}(\cdot|x)}\left[\log p_{\boldsymbol{\theta}}(s_1^{(i)} \mid u_1^{(i)})\right] &= \sum_{k=1}^{C} \alpha_{1,k}^{(i)}(x)\left(-\frac{1}{2}\log 2\pi \bar{\psi}_k^{(i)} - \frac{1}{2}(\bar{\psi}_k^{(i)})^{-1}\mathbb{E}_{q_{\boldsymbol{\varphi}}(\cdot|x)}\left[(s_1^{(i)} - \bar{b}_k^{(i)})^2\right]\right) \\
&= -\frac{1}{2}\sum_{k=1}^{C} \alpha_{1,k}^{(i)}(x)\left(\log 2\pi \bar{\psi}_k^{(i)} + (\bar{\psi}_k^{(i)})^{-1}\left(\Sigma_1^{(i,i)}(x) + (m^{(i)}(x) - \bar{b}_k^{(i)})^2\right)\right) \, .
\end{aligned}
\tag{47}
$$

Grouping previous results together yields the ELBO explicit form

$$
\begin{aligned}
\mathcal{L}(x;\boldsymbol{\theta},\boldsymbol{\varphi}) = & \sum_{t=1}^{T} 1_K^\top \Big( x_t \odot \big(\Gamma\mu_t(x)\big) - \exp\Big(\Gamma\mu_t(x) + \frac{1}{2}\mathrm{diag}\big(\Gamma\Sigma_t(x)\Gamma^\top\big)\Big) - \log(x_t!) \Big) \\
& - \sum_{i=1}^{d} \left( \sum_{k=1}^{C} \nu_k^{(i)}(x) \log \frac{\nu_k^{(i)}(x)}{\pi_k^{(i)}} + \sum_{t=1}^{T-1}\sum_{k=1}^{C}\sum_{\ell=1}^{C} \tau_{t+1,k\ell}^{(i)}(x)\alpha_{t,\ell}^{(i)}(x)\log\frac{\tau_{t+1,k\ell}^{(i)}(x)}{a_{k\ell}^{(i)}} \right) \\
& + \frac{Td}{2}\log 2\pi e + \frac{1}{2}\sum_{t=1}^{T}\log\det\tilde{\psi}_t(x) \\
& - \frac{1}{2}\sum_{i=1}^{d}\sum_{k=1}^{C}\alpha_{1,k}^{(i)}(x)\Big(\log 2\pi\bar{\psi}_k^{(i)} + (\bar{\psi}_k^{(i)})^{-1}\big(\Sigma_1^{(i,i)}(x) + (m^{(i)}(x) - \bar{b}_k^{(i)})^2\big)\Big) \\
& - \frac{1}{2}\sum_{i=1}^{d}\sum_{t=1}^{T-1}\sum_{k=1}^{C}\alpha_{t+1,k}^{(i)}(x)\Big(\log 2\pi\psi_k^{(i)} + (\psi_k^{(i)})^{-1}\big((\mu_{t+1}^{(i)}(x) - B_k^{(i)}\mu_t^{(i)}(x) - b_k^{(i)})^2 \\
& \qquad\qquad\qquad + \Sigma_{t+1}^{(i,i)}(x) + B_k^{(i)}(B_k^{(i)} - 2\tilde{B}_{t+1}^{(i)})\Sigma_t^{(i,i)}(x)\big)\Big) .
\end{aligned}
\tag{48}
$$

For computational efficiency, we report a tensorized form of the ELBO equivalent to the above which is given by

$$
\begin{aligned}
\mathcal{L}(x;\boldsymbol{\theta},\boldsymbol{\varphi}) = & \sum_{t=1}^{T} 1_K^\top \Big( x_t \odot \big(\Gamma\mu_t(x)\big) - \exp\Big(\Gamma\mu_t(x) + \frac{1}{2}\mathrm{diag}\big(\Gamma\Sigma_t(x)\Gamma^\top\big)\Big) - \log(x_t!) \Big) \\
& - 1_d^\top \left( \nu(x) \odot \log(\nu(x) \oslash \pi) \right) 1_C + \sum_{t=1}^{T-1} 1_d^\top \left( \tau_{t+1}(x) \odot \alpha_t^*(x) \odot \log(\tau_{t+1}(x) \oslash A) \right) 1_{C^2} \\
& + \frac{Td}{2}\log 2\pi e + \frac{1}{2}\sum_{t=1}^{T}\log\det\tilde{\psi}_t(x) - \frac{1}{2}1_d^\top \left( \alpha_1(x) \odot (\log 2\pi\bar{\psi} + (\sigma_1^*(x) + (m^*(x) - \bar{b})^2) \oslash \bar{\psi}) \right) 1_C \\
& - \frac{1}{2}\sum_{t=1}^{T-1} 1_d^\top \Big( \alpha_{t+1}(x) \odot \Big( \log 2\pi\psi + \big((\mu_{t+1}^*(x) - B \odot \mu_t^*(x) - b)^2 \\
& \qquad\qquad\qquad + \sigma_{t+1}^*(x) + B \odot (B - 2\tilde{B}_{t+1}^*) \odot \sigma_t^*(x)\big) \oslash \psi \Big) \Big) 1_C ,
\end{aligned}
\tag{49}
$$

where $\sigma_t(x) \in \mathbb{R}^d$ is the diagonal of $\Sigma_t(x)$.

## C. CAVI update for the variational latent switching states

While joint parametric inference is most employed in variational frameworks, the choice of the parametric family is often arbitrary simple and can miss on structural modeling such as model conjugacy. Besides, the parametric forms often come with deep neural parameterization which entail significant engineering challenges in practice. In the context of mean-field variational approximations defined by

$$
q_{\boldsymbol{\varphi}}(z \mid x) = \prod_{i=1}^{d} q_{\boldsymbol{\varphi}}^{(i)}(z^{(i)} \mid x) ,
\tag{50}
$$

coordinate ascent variational inference (CAVI) offers a grounded framework to alleviate these issues, relying on maximizing a coordinate-wise ELBO, such that for coordinate $i$ at iteration $h$:

$$
\mathcal{L}_h(x;\boldsymbol{\theta},\boldsymbol{\varphi}) = \mathbb{E}_{q_{\boldsymbol{\varphi}}^{(i)}(\cdot|x)}\left[ \mathbb{E}_{q_{\boldsymbol{\varphi}^{(h)}}^{(-i)}(\cdot|x)}[\log p_{\boldsymbol{\theta}}(x,z)] \right] + \mathbb{E}_{q_{\boldsymbol{\varphi}}^{(i)}(\cdot|x)}\left[ \log q_{\boldsymbol{\varphi}}(z^{(i)} \mid x) \right] ,
\tag{51}
$$

where $\boldsymbol{\varphi}^{(h)}$ are the parameters of the previous iterate of the coordinate-ascent, and $q_{\boldsymbol{\varphi}}^{(-i)}(\cdot \mid x) = \prod_{j\neq i}^{d} q_{\boldsymbol{\varphi}}^{(j)}(\cdot \mid x)$. When considering variational approximations from the exponential family, Johnson et al. (2016) provides optimal CAVI updates

for the variational proxy given by

$$q_{\boldsymbol{\varphi}}^{(i)}(z^{(i)} \mid x) \ \propto \ \exp\left\{\mathbb{E}_{q_{\boldsymbol{\varphi}^{(h)}}^{(-i)}(\cdot\mid x)}[\log p_{\boldsymbol{\theta}}(x,z)]\right\} , \tag{52}$$

which can be explicated under conjugacy conditions between the exponential family and prior terms. In the context of ARPLN-ICA inference, we consider a mean field decomposition across components $i$, and between latent switching states and latent sources in Eq. (5). Additionally, since $u_{1:T}^{(i)}$ only affects $s_{1:T}^{(i)}$ under the model prior, conjugacy is satisfied by definition of $p_{\boldsymbol{\theta}}(u_{1:T}^{(i)})$ and $p_{\boldsymbol{\theta}}(s_{1:T}^{(i)} \mid u_{1:T}^{(i)})$, and the CAVI update depends solely on $q_{\boldsymbol{\varphi}}(s_{1:T}^{(i)} \mid x)$. Thus, denoting by $q_{\boldsymbol{\varphi},s}^{(i)}(s_{1:T}^{(i)} \mid x) = q_{\boldsymbol{\varphi}}(s_{1:T}^{(i)} \mid x)$, the CAVI update for the variational approximation $q_{\boldsymbol{\varphi}}(u_{1:T}^{(i)} \mid x)$ can be explicated as

$$q_{\boldsymbol{\varphi}}(u_{1:T}^{(i)} \mid x) \ \propto \ \exp \mathbb{E}_{q_{\boldsymbol{\varphi},s}^{(i)}(\cdot\mid x)}\left[\log p_{\boldsymbol{\theta}}(s_1^{(i)} \mid u_1^{(i)}) + \log p_{\boldsymbol{\theta}}(u_1^{(i)}) + \sum_{t=1}^{T-1} \log p_{\boldsymbol{\theta}}(s_{t+1}^{(i)} \mid s_t^{(i)}, u_{t+1}^{(i)}) + \log p_{\boldsymbol{\theta}}(u_{t+1}^{(i)} \mid u_t^{(i)})\right]$$

$$\propto \ \pi_{u_1^{(i)}}^{(i)} e_1^{(i)}(u_1^{(i)}) \cdot \prod_{t=1}^{T-1} a_{u_t^{(i)},u_{t+1}^{(i)}}^{(i)} e_{t+1}^{(i)}(u_{t+1}^{(i)}) , \tag{53}$$

where the evidence terms are computed using the marginal distribution of the variational latent sources described by Eq. (39) and the auto-correlation of Eq. (46), yielding

$$e_1^{(i)}(k) = (\bar{\psi}_k^{(i)})^{-1/2} \exp\left(-\frac{\Sigma_1^{(i,i)}(x) + (\mu_1^{(i)}(x) - \bar{b}_k^{(i)})^2}{2\bar{\psi}_k^{(i)}}\right) ,$$

$$e_{t+1}^{(i)}(k) = (\psi_k^{(i)})^{-1/2} \exp\left(-\frac{\Sigma_{t+1}^{(i,i)}(x) + (B_k^{(i)})^2\Sigma_t^{(i,i)}(x) + (\mu_{t+1}^{(i)}(x) - B_k^{(i)}\mu_t^{(i)}(x) - b_k^{(i)})^2 - 2B_k^{(i)}\Sigma_{t,t+1}^{(i,i)}(x)}{2\psi_k^{(i)}}\right) . \tag{54}$$

As such, the latent regimes distribution is component-wise independent Markov chains which are optimally derived in closed-form in one CAVI step from the sources variational proxy $\big(q_{\boldsymbol{\varphi}}(s^{(i)} \mid x)\big)_{1\le i\le d}$.

## D. Closed-form updates of prior parameters

Inference in ARPLN-ICA requires estimating both the model parameters $\boldsymbol{\theta}$ (prior and generative parameters) and the variational parameters $\boldsymbol{\varphi}$. The variational distribution over sources is fitted via stochastic optimization due to the deep amortized parameterization of the dynamics. In contrast, Appendix C shows that, conditional on the source variational factors, the variational distribution over the latent switching labels admits an optimal update within the exponential family. Accordingly, within a coordinate-ascent view of ELBO maximization, we can treat $\boldsymbol{\varphi}$ as fixed and update $\boldsymbol{\theta}$ by maximizing the ELBO conditional on the current variational approximation. With the exception of the mixing function, all ARPLN-ICA model parameters admit closed-form updates. To decline the different expressions, we use the variational marginal kernel parameters of the sources in Eq. (39) and the variational marginal of the latent switching states in Eq. (41).

- Prior latent switching parameters:

$$(\pi_k^{(i)})^* = \frac{1}{|\mathcal{D}|} \sum_{x\in\mathcal{D}} \nu_k^{(i)}(x) ,$$

$$(a_{k\ell}^{(i)})^* = \frac{\sum_{x\in\mathcal{D}} \sum_{t=1}^{T-1} \tau_{t+1,k\ell}^{(i)}(x)\alpha_{t,\ell}^{(i)}(x)}{\sum_{x\in\mathcal{D}} \sum_{t=1}^{T-1} \alpha_{t,\ell}^{(i)}(x)} . \tag{55}$$

- Initialisation of the latent sources prior parameters:

$$(\bar{b}_k^{(i)})^* = \frac{\sum_{x\in\mathcal{D}} \alpha_{1,k}^{(i)}(x)m^{(i)}(x)}{\sum_{x\in\mathcal{D}} \alpha_{1,k}^{(i)}(x)} ,$$

$$(\bar{\psi}_k^{(i)})^* = \frac{\sum_{x\in\mathcal{D}} \alpha_{1,k}^{(i)}(x)\big(\Sigma_1^{(i,i)}(x) + (m^{(i)}(x) - \bar{b}_k^{(i)})^2\big)}{\sum_{x\in\mathcal{D}} \alpha_{1,k}^{(i)}(x)} . \tag{56}$$

- Auto-regressive sources prior dynamic parameters:

$$
(B_k^{(i)})^* = \frac{\sum_{x \in \mathcal{D}} \sum_{t=1}^{T-1} \alpha_{t+1,k}^{(i)}(x) \left( \tilde{B}_{t+1}^{(i)}(x) \Sigma_t^{(i,i)}(x) + \mu_t^{(i)}(x)(\mu_{t+1}^{(i)}(x) - b_k^{(i)}) \right)}{\sum_{x \in \mathcal{D}} \sum_{t=1}^{T-1} \alpha_{t+1,k}^{(i)}(x) \left( (\mu_t^{(i)}(x))^2 + \Sigma_t^{(i,i)}(x) \right)} ,
$$

$$
(b_k^{(i)})^* = \frac{\sum_{x \in \mathcal{D}} \sum_{t=1}^{T-1} \alpha_{t+1,k}^{(i)}(x)(\mu_{t+1}^{(i)}(x) - B_k^{(i)} \mu_t^{(i)}(x))}{\sum_{x \in \mathcal{D}} \sum_{t=1}^{T-1} \alpha_{t+1,k}^{(i)}(x)} ,
$$

$$
(\psi_k^{(i)})^* = \Big[ \sum_{x \in \mathcal{D}} \sum_{t=1}^{T-1} \alpha_{t+1,k}^{(i)}(x)\big((\mu_{t+1}^{(i)}(x) - B_k^{(i)} \mu_t^{(i)}(x) - b_k^{(i)})^2 + \Sigma_{t+1}^{(i,i)}(x)
$$

$$
+ B_k^{(i)}(B_k^{(i)} - 2\tilde{B}_{t+1}^{(i)}(x))\Sigma_t^{(i,i)}(x))\Big] \Big/ \sum_{x \in \mathcal{D}} \sum_{t=1}^{T-1} \alpha_{t+1,k}^{(i)}(x) .
$$

(57)

Note that the updates above can be fully tensorized, ensuring efficient implementations. While $\Gamma$ does not admit a closed-form update, the ELBO is convex with respect to the columns of $\Gamma$, which facilitates stochastic optimization of this parameter during our training procedure.

## E. Simulated studies material

### E.1. Variational inference with mean field approximation

In ARPLN-ICA inference procedure, we propose a variational approximation which captures part of the model's structure in Eq. (5). Specifically, we exploit the Markov structure of the true posterior to parameterize the variational distribution over latent sources via a filtering factorization in Eq. (6), and leverage conditional conjugacy to derive an optimal exponential-family proxy for the latent switching states (Johnson et al., 2016). A common alternative is the variational mean field approximation (Blei et al., 2017), which ignores posterior dependencies by assuming coordinate-wise independence. Applied to the latent sources, the mean field approximation yields a time-wise and component-wise factorized proxy,

$$
q_{\boldsymbol{\varphi}}^{\mathrm{MF}}(s_{1:T} \mid x) = \prod_{t=1}^{T} \prod_{i=1}^{d} q_{\boldsymbol{\varphi}}^{\mathrm{MF}}(s_t^{(i)} \mid x) .
$$

(58)

Despite its limited structure, the mean-field approximation can exhibit beneficial regularization effects which can help mitigating amortization shortcuts and mode collapse relative to richer variational families (Shu et al., 2018). Importantly, we do not impose a mean-field factorization on the switching states, as conditional conjugacy still enables CAVI updates for the latent switches, and further factorization would only degrade the joint approximation. In our parameterization, applying mean-field to the sources is equivalent to setting the auto-regressive coefficient in Eq. (6) to zero. Accordingly, when comparing the proposed structured AR proxy of Eq. (6) to the mean-field approximation, we use the same neural parameterization and implement a targeted ablation that removes only the auto-regressive term, ensuring a controlled comparison between the two approximations.

### E.2. Linear ICA baselines

Blind source separation and ICA are commonly formulated through an instantaneous linear mixing model

$$
x_t = As_t + \varepsilon_t ,
$$

(59)

where $A \in \mathbb{R}^{K \times d}$ is unknown, $s_t \in \mathbb{R}^d$ are latent sources, and $\varepsilon_t$ captures noise. Methods differ mainly by the structural assumptions used to identify $A$: second-order source separation exploits covariance and auto-correlation structure in multivariate time series (Pan et al., 2022), while likelihood and contrast-based ICA rely on non-Gaussianity and independence of the components (Hyvärinen & Oja, 2000).

UWEDGEICA is a second-order estimator implemented in COROICA (Pfister et al., 2019), based on approximate joint diagonalization via the UWEDGE routine (Tichavsky & Yeredor, 2008). In our multi-trajectory setting, temporal data are globally centered and PCA-whitened to the target dimension $d$ (see Hyvärinen & Oja (2000)). We then run UWEDGEICA

with non-overlapping windows of length 10 and time lags $\tau \in \{1, \ldots, 9\}$. The method returns an unmixing estimator $A^\dagger$, from which we recover a mixing estimate $A$ by pseudo-inversion.

PICARD is a modern maximum-likelihood ICA solver designed for fast, robust optimization under model misspecification (Ablin et al., 2018). Contrary to second-order temporal methods, it does not leverage temporal dependence and is therefore fit on time-flattened observations, using default parameters from the `python-picard` package of the original authors.

In the simulation setting, both baselines are applied to log-transformed counts $y_t = \log(x_t + \varepsilon)$ with $\varepsilon = 0.5$. Under our ARPLN-ICA model, $x_t \mid s_t \sim \mathcal{P}(\exp(\Gamma s_t))$, so that $\mathbb{E}[x_t \mid s_t] = \exp(\Gamma s_t)$ and hence $\log \mathbb{E}[x_t \mid s_t] = \Gamma s_t$. Therefore $y_t$ can be viewed as a noisy, approximately linear instantaneous mixture of the form $y_t \approx \Gamma s_t + \eta_t$, where $\eta_t$ is heteroskedastic and non-Gaussian due to Poisson sampling and the log transform. Consequently, provided this approximation is sufficiently accurate, we expect linear ICA fitted on $y_t$ to recover a mixing approximately aligned with $\Gamma$.

### E.3. Effect of $\Gamma$ on recovery: study design

**Simulation setting.** To assess finite-sample recovery of $\Gamma$ under our inference procedure, we consider three simulation scenarios that vary the structure of the mixing matrix. Across all scenarios, we fix $K = 12$, $T = 20$, and $d = 5$, and sample $n = 150$ trajectories from an oracle model with parameters and training set drawn once at random with fixed seed. The three regimes differ only through the construction of $\Gamma$, obtained from Gaussian initializations followed by scenario-specific offsets and rescalings that control column colinearity and component expressivity (see `simulation_study` notebook on our provided GitHub repository).

**Scenarios' description.** The first two regimes study the effect of column colinearity since highly aligned columns are expected to be more difficult to disentangle and may hinder identification. We quantify column coherence through the absolute Gram matrix $|\Gamma^\top \Gamma| \in \mathbb{R}^{d \times d}$, where entry $(i, j)$ equals $|\langle \gamma_i, \gamma_j \rangle|$ with $\Gamma = [\gamma_i]_{1 \le i \le d}$, as illustrated on Figure 8 across scenarios. In the *moderate-coherence* and *low-excitation* regimes, the maximum pairwise coherence is $0.65$ and $0.51$, respectively, whereas the *high-coherence* regime exhibits substantially stronger alignment (maximum $0.87$), with each column displaying moderate to high coherence with at least one other component.

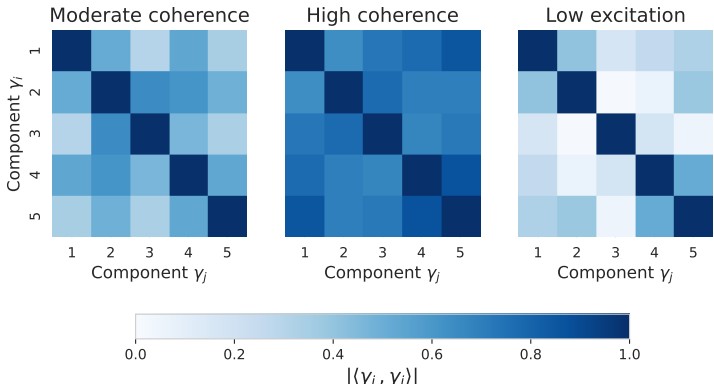

*Figure 8.* Absolute Gram matrix per scenario. Off-diagonal entries quantify pairwise column coherence: the moderate-coherence regime reaches $0.65$, the high-coherence regime $0.87$, and the low-excitation regime $0.51$.

Beyond coherence, we vary the *excitation* of the latent sources, that is, the extent to which components contribute to the Poisson emission through the log-intensities $z_t^{(k)} = \sum_{j=1}^d \Gamma_j^{(k)} s_t^{(j)}$. In the moderate- and high-coherence regimes, most components are expressed in the observations, with generally positive contributions in log-intensity In contrast, the *low-excitation* regime is designed so that many components induce only weak log-intensity signals. Figure 9 reports time-averaged log-intensities (with standard deviation) and shows that low excitation translates into substantially sparser observations, with $\approx 31\%$ zeros on average over time, compared less than $0.1\%$ in the other regimes.

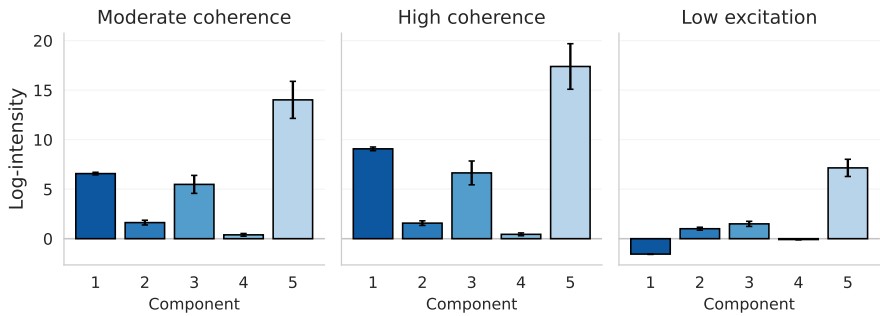

*Figure 9.* Time-averaged Poisson log-intensity per component, with standard deviation. The low-excitation regime yields smaller log-intensities and a substantially higher fraction of zeros in the observations ($\approx 31\%$).

**Training procedure.** For each scenario, we train ARPLN-ICA models using the same hyperparameters as the oracle, but with randomly initialized parameters. When offsets are used to generate the oracle data, we provide the corresponding offset specification to the fitted models to ensure a matched recovery setting. To quantify variability due to optimization and initialization, we repeat training 10 times with different random seeds. For the variational proxy, we adopt the deep amortized parameterization shown in Figure 10. We encode each sequence into a latent representation $\tilde{x}_t$ of dimension 4 using a two-layer GRU, followed by a two-layer feed-forward network. The recognition network is a two-layer ReLU network with hidden dimensions $[16, 8]$, and each parametric head is implemented as a two-layer ReLU network with hidden width 8. No baseline fixed-effect is considered in these simulation settings. Model optimization is conducted using AdamW in PyTorch's framework (Paszke et al., 2019) for 800 epochs with learning rate $10^{-3}$, weight decay of $10^{-4}$, gradient clipping at norm 5, and cosine annealing scheduling at $T_{\max} = 800$. Convergence during training is checked with relative tolerance of $10^{-4}$ over the 10 last values of the ELBO.

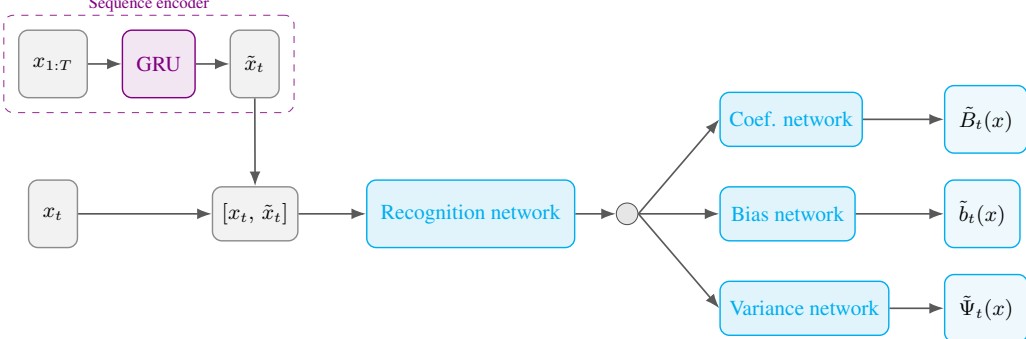

*Figure 10.* Auto-regressive variational approximation with neural parameterization. The full sequence $x_{1:T}$ is first processed by a gated recurrent unit (GRU), producing a fixed-dimensional embedding $\tilde{x}_t$ at each time step. This embedding is concatenated with the current observation $x_t$ and passed to a shared recognition network (feature extractor) implemented as a fully connected ReLU network. The resulting features are then routed to three head networks, each one outputting a variational parameter for time steps $t \geq 1$. At $t = 0$, we use the same architecture but remove the coefficient head $\tilde{B}$. All heads share the same architecture, and all networks use ReLU activations. To stabilize training, counts are normalized to proportions. When provided, offset information are concatenated to the input $x_t$.

### E.4. Quantitative comparison of inferred $\Gamma$

**Mixing matching.** Based on Corollary 3.2, the mixing matrix $\Gamma$ is identifiable only up to a permutation of its columns and component-wise signed scaling. During inference, we constrain each column of $\Gamma$ to lie in the unit $\ell_2$-ball, which removes scale indeterminacy and leaves only column permutations and sign flips as residual invariances. Consequently, to compare an estimated $\Gamma$ to an oracle column-normalized matrix $\Gamma^*$, we first align the columns and signs of $\Gamma$ to those of $\Gamma^*$ prior to computing similarity scores. To this end, we implement a brute-force alignment procedure that enumerates all signed permutations and selects the permutation maximizing colinearity with the columns $\Gamma^*$: the best permutation is obtained as the maximizer of the sum of the absolute scalar products between the permuted columns of $\Gamma$ and the fixed columns of $\Gamma^*$.

**Mixing similarity.** We quantify similarity between two aligned mixing matrices by measuring column-wise similarity under the remaining sign indeterminacy. For vectors $v, w \in \mathbb{R}^d$, a matching metric is given by the cosine similarity as

$$S_{\cos}(v, w) = \frac{\langle v, w \rangle}{\|v\|_2 \|w\|_2} . \tag{60}$$

Since mixing columns are $\ell_2$-normalized during inference, the cosine similarity exactly coincides with the inner product. In particular, the absolute cosine similarity assesses the components similarity under sign invariance, such that 1 reflects equivalent components while 0 reflects orthogonal components, making it a suitable metric to assess mixing similarity.

### E.5. Scalability to increasing effective training set size

When fitting state-space models, computational cost notably depends on both the number of independent trajectories $n$ and the sequence length $T$. In this experiment, we study runtime scaling with $n$ in the moderate-coherence simulation setting for three time series lengths $T \in \{10, 20, 40\}$. Figure 11 reports training time (seconds per epoch) under full-batch optimization with same hyperparameterization. Across all values of $T$, the per-epoch runtime is approximately flat for small to moderate $n$, before sharply increasing beyond a threshold (around $n \approx 300$ in this implementation), with longer sequences involving a higher cost at fixed $n$. For the configuration used in Section 4 ($T = 20$) and 800 epochs of training, this corresponds to roughly 2 minutes of wall-clock time for $n \leq 500$ and up to 6 minutes for $n = 3{,}000$ on an RTX 4080S (16 GB). In the same setting, we typically observe diminishing returns in recovery metrics beyond $n \approx 100$ (see Figure 3), suggesting that, for this problem size, substantially larger $n$ may increase compute without measurable gains.

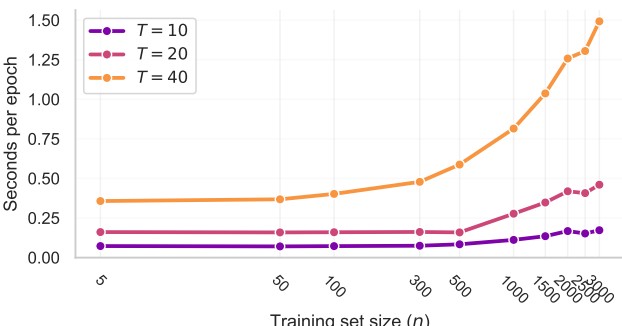

*Figure 11.* Training speed (seconds per epoch) as a function of training size $n$ in the moderate-coherence scenario with varying time series length $T$ ($K = 12$, $d = 5$). Experiments run on CUDA (RTX 4080S 16 GB) using full-batch optimization and the training setup of Appendix E.3. All models are trained with the same architecture and optimization procedure, starting with the same random state.

## F. Clostridium Difficile ICA numerical analysis

### F.1. Design of the analysis

The gnotobiotic mice dataset of (Bucci et al., 2016) comprises $n = 5$ longitudinal trajectories, each with $T = 26$ time points over $K = 13$ microbial taxa. Although this yields $n \times T = 130$ effective observations, learning the underlying dynamics remains challenging due to the small number of independent trajectories. In this context, we adopt a compact and regularized parameterization throughout the experiments. Fixing $d = 4$, $C = 2$ for the prior model parameters, we use the encoder architecture of Figure 10 with an sequence embedding dimension of 10 obtain through a 3-layer GRU, followed by a single linear layer before concatenation with the current observation. The concatenated features are passed into a 2-layer recognition network (fully connected) with hidden sizes $[32, 16]$, and each variational head is a 2-layer network with hidden size 16. Additionally, we account for offset noise and fixed-effects as in Section 3.4, using the logsum offsets described in Appendix G.4. We perform the optimization using AdamW with learning rate $10^{-3}$ and weight decay $10^{-3}$ for 800 epochs, applying gradient clipping at 5, and using cosine annealing with $T_{\max} = 720$. Convergence is monitored through out 10 epochs at relative tolerance $10^{-3}$. Figure 12 reports ELBO optimization trajectories in log-scale across the leave-one-out cross-validation folds, indicating stable convergence.

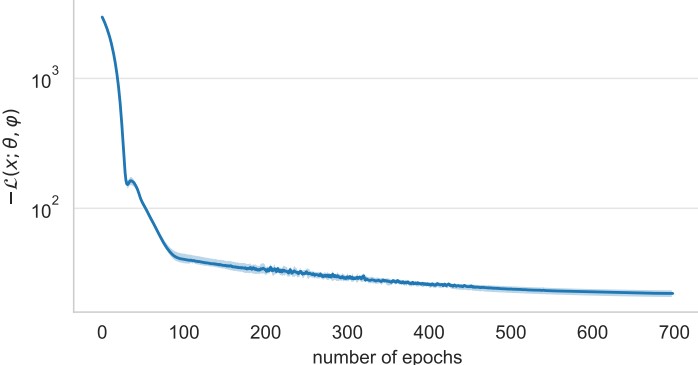

*Figure 12.* Mean negative ELBO over training epochs in log scale, with standard deviation across LOO-CV folds.

### F.2. Empirical switching distribution for dataset-level perturbations

To summarize perturbation dynamics at the dataset level, we define an *empirical switching distribution* by uniformly averaging the variational posteriors across samples, in the spirit of a VAMP distribution (Tomczak & Welling, 2018),

$$\bar{q}_{\boldsymbol{\varphi}}(u_{1:T}) = \frac{1}{|\mathcal{D}|} \sum_{x \in \mathcal{D}} q_{\boldsymbol{\varphi}}(u_{1:T} \mid x) . \tag{61}$$

Each term in the mixture is a discrete Markov chain, which allows us to compute mixture marginals and transition kernels in closed form as

$$\bar{q}_{\boldsymbol{\varphi}}(u_t^{(i)} = k) = \bar{\alpha}_t^{(i)}(k) = \frac{1}{|\mathcal{D}|} \sum_{x \in \mathcal{D}} \alpha_t^{(i)}(k, x) ,$$

$$\bar{q}_{\boldsymbol{\varphi}}(u_{t+1}^{(i)} = k \mid u_t^{(i)} = \ell) = \bar{\tau}_{t+1,k\ell}^{(i)} = \frac{\frac{1}{|\mathcal{D}|} \sum_{x \in \mathcal{D}} \tau_{t+1,k\ell}^{(i)}(x) \, \alpha_t^{(i)}(\ell, x)}{\bar{\alpha}_t^{(i)}(\ell)} , \tag{62}$$

where $\alpha_t^{(i)}(k, x) = q_{\boldsymbol{\varphi}}(u_t^{(i)} = k \mid x)$ and $\tau_{t+1,k\ell}^{(i)}(x) = q_{\boldsymbol{\varphi}}(u_{t+1}^{(i)} = k \mid u_t^{(i)} = \ell, x)$. Figure 4 reports the state 0 marginal of $\bar{q}_{\boldsymbol{\varphi}}$ over time, providing an aggregated view of regime switching across the dataset. For per-mouse trajectories, Figure 13 shows the marginal probability of state 0 under each mixture component. Although individual mice exhibit trajectory-specific variations, the dataset-level mean marginal provides a representative summary of regime occupancy.

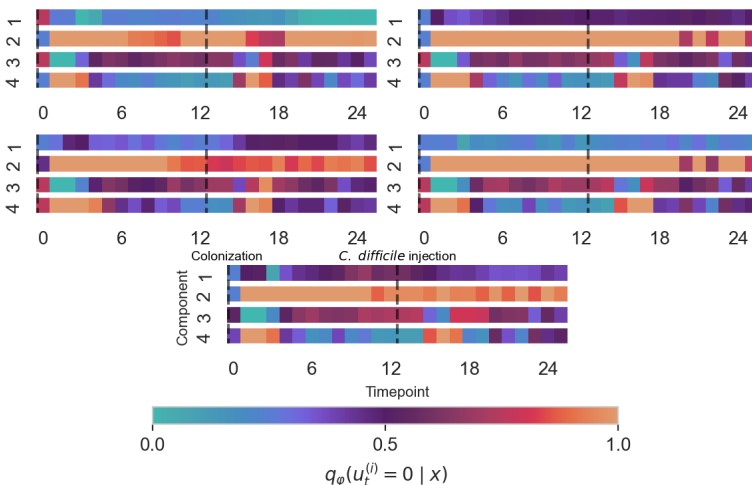

*Figure 13.* Marginal probability of being in latent state 0 for each component under the variational approximation, computed on test samples across LOO-CV. Switching states are aligned component-wise with the first LOO mice to mitigate invariance between LOO.

### F.3. Components stability and medoid mixing.

**Inference stability.** We assess the stability of ARPLN-ICA on the gnotobiotic mice dataset using leave-one-out cross-validation (LOO-CV), which yields 5 estimated mixing matrices $\Gamma$ (one per split). Figure 14 reports pairwise absolute cosine similarities, averaged across columns, together with per-column average similarities and $95\%$ confidence intervals. The inferred mixings are highly consistent across folds, with mean similarity exceeding $98\%$, supporting fold-wise comparability of the recovered components.

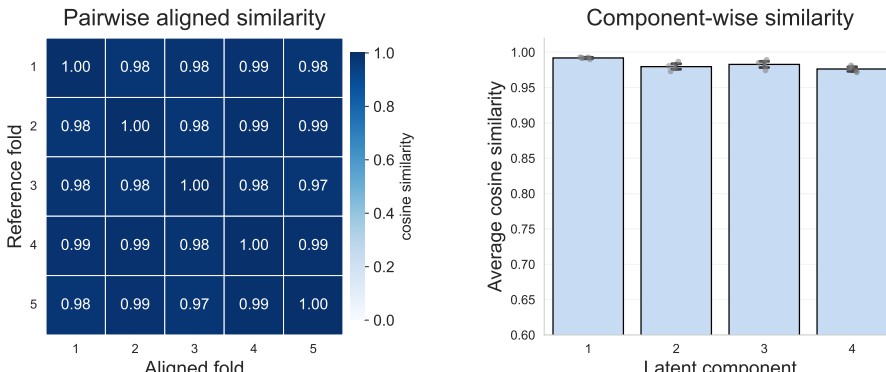

*Figure 14.* Stability of the learned mixing matrix $\Gamma$ across leave-one-out folds. (Left) Pairwise cosine similarity between $\Gamma$ estimates after aligning fold $j$ to fold $i$ (permutation/sign invariances). (Right) Component-wise stability, computed as the mean absolute cosine similarity of each aligned column, aggregated within each reference fold and summarized across folds, with $95\%$ confidence interval and similarity per fold indicated by grey dots.

**Medoid mixing.** To enable dataset-level interpretation of the learned mixing across LOO CV, we select a single reference matrix to which all other estimates can be aligned. We define this *medoid mixing* $\bar{\Gamma}$ as the mixing matrix whose average similarity to the other fold-specific estimates is maximal. In Figure 14, this corresponds to the fold that maximizes the column-averaged pairwise similarities, which occurs for the 4-th split. We use $\bar{\Gamma}$ as an anchor to align the other inferred mixings across folds, yielding a consistent dataset-level representation of components. To quantify variability around the medoid on a per-component basis, we consider the empirical variance estimator

$$\hat{\mathbb{V}}(\bar{\gamma}_i) = \frac{1}{n_{\mathrm{CV}} - 1} \sum_{k=1}^{n_{\mathrm{CV}}} \left( \hat{\gamma}_{k,i} - \bar{\gamma}_i \right)^2 , \tag{63}$$

where $n_{\mathrm{CV}}$ is the number of splits, $\bar{\gamma}_i$ denotes the $i$-th column of the medoid mixing, and $\hat{\gamma}_{k,i}$ is the corresponding column inferred on split $k$ after alignment.

### F.4. Effect of the number of components in C. difficile analysis

**Model selection procedure.** Choosing the number of latent components $d$ is a key design choice in ICA. In practice, there is no universally optimal criterion, as $d$ is typically selected based on task-dependent evaluations using either reconstruction or predictive performance under cross-validation, likelihood-based criteria, or stability-based diagnostics (see Himberg et al. (2004); Hu et al. (2020); Yi et al. (2024) and the references therein). Although likelihood-based criteria would be natural for our generative formulation, the variational inference procedure introduces complexities: the marginal likelihood is intractable and the amortized variational parameterization blurs the notion of effective model complexity, complicating AIC/BIC-style estimators definition. Therefore, we propose an optimal $d$ based on out-of-sample quantitative metrics (see Appendix H) computed from reconstructed trajectories of held-out mice in the LOO-CV. Figure 15 reports these metrics as a function of $d$ and suggests a performance optimum around $d = 7$. Although $d = 4$ is not optimal, Figure 16 shows that it still generally recovers the microbial pattern through reconstruction on the test set.

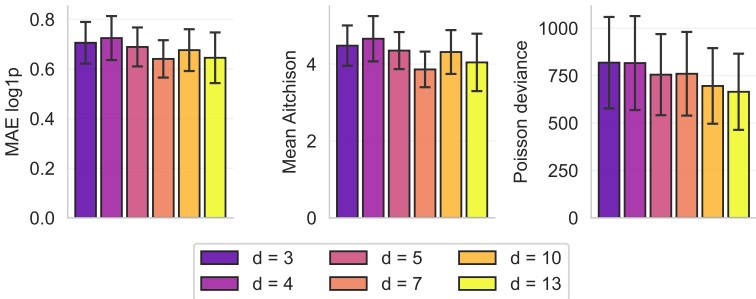

*Figure 15.* Average reconstruction performance on test mice from LOO CV, with increasing dimensionality of the latent sources. Barplots show the mean and standard deviation across LOO splits of the scoring value averaged along the test mice trajectory. All methods are evaluated with the same parameterization, with varying $d$.

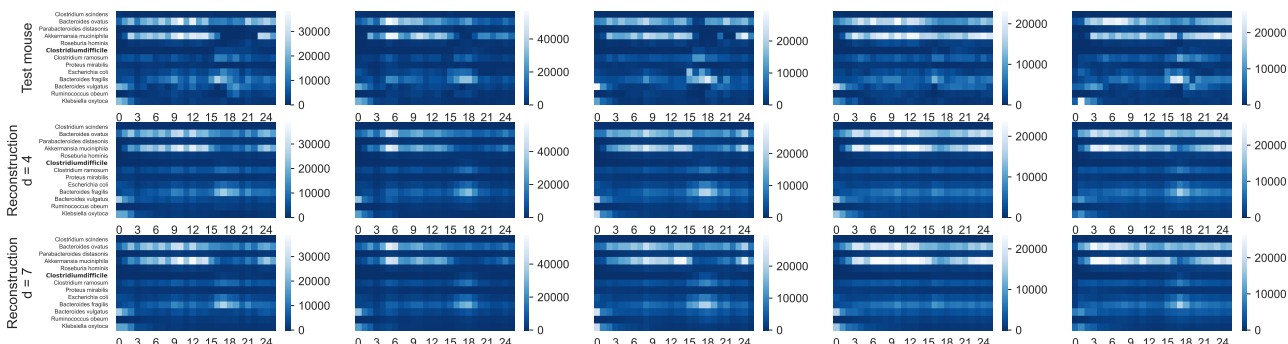

*Figure 16.* Test mice and reconstruction with ARPLN-ICA at $d \in [4, 7]$ from LOO CV. Reconstructed trajectories generally capture the patterns in microbial taxa abundance in the test set, with sharper reconstruction for $d = 7$, consistent with the quantitative benchmark.

**Biological interpretation at $d = 7$.** For $d = 4$, Figure 5 identifies component 4 as strongly responsive to the perturbation, with a co-variation pattern that contrasts opportunistic pathobionts against known protective taxa. At $d = 7$, we recover an analogous signal: Figure 17 highlights component 5 as exhibiting a sharp perturbation response, while Figure 18 shows that the loadings of *C. difficile* aligns with the same opportunistic pathobionts as with $d = 4$: *E. coli*, *B. fragilis*, and *C. ramosum*. In contrast, *A. muciniphila* loads in the opposite direction alongside *P. distasonis* and *R. hominis*, which are commonly regarded as commensal, healthy-gut-associated species. Overall, although $d$ is optimized using out-of-sample reconstruction metrics, the biological interpretation is stable across the tested dimensionalities, with further complexities illustrated as the dimension increases.

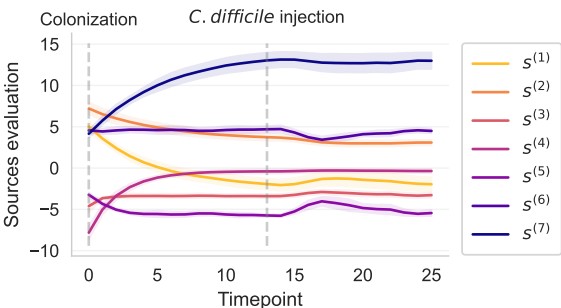

*Figure 17.* Average encoded latent sources with $d = 7$ on the gnotobiotic mice dataset. The evolution of the latent sources through time is computed for each test mice in the LOO, aligned with the medoid mixing, with standard deviation.

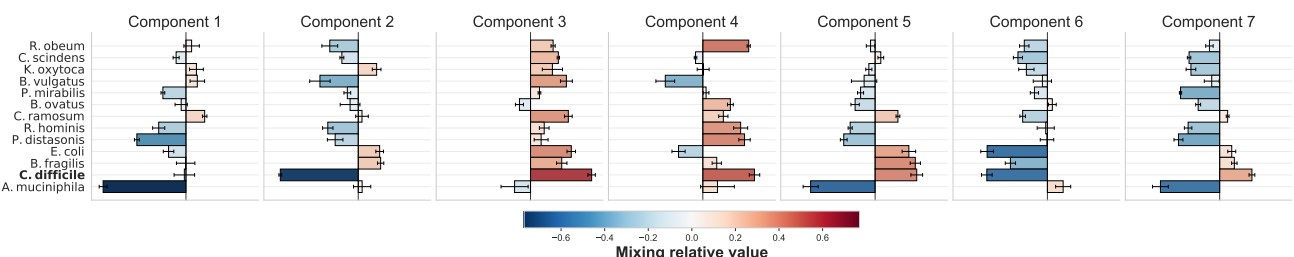

*Figure 18.* Mixing loading per taxa and sources with $d = 7$ on the gnotobiotic mice dataset. Taxa loading per component (column) is computed from the medoid mixing across LOO CV, with standard deviation to the medoid. The 5-th component displays a similar role as the 4-th component of the case $d = 4$ in Figure 5.

## G. Forecasting with empirical inhomogeneous switching

### G.1. Online forecasting variational proxy

Given count observations $x_{1:T}$, a natural practical goal is forecasting, that is, characterizing the filtering distribution $p_{\boldsymbol{\theta}}(x_{T+1} \mid x_{1:T})$. In a typical hidden Markov model with latent variables $z_{1:T}$, the filtering distribution satisfies

$$p_{\boldsymbol{\theta}}(x_{T+1} \mid x_{1:T}) = \int p_{\boldsymbol{\theta}}(x_{T+1} \mid z_{T+1}) \, p_{\boldsymbol{\theta}}(z_{T+1} \mid z_T) \, p_{\boldsymbol{\theta}}(z_T \mid x_{1:T}) \, dz_{T:T+1} \ . \tag{64}$$

In the context of ARPLN-ICA, we can approximate this distribution using the variational posterior. In particular, by adopting a filtering variational family by restricting the conditioning at time $t$ to $x_{1:t}$ in Eq. (5), we obtain a fully online procedure that supports sequential state estimation and counts prediction for arbitrary horizons. This variational proxy trades off information from future observations for scalability and streaming compatibility, as the resulting predictive distributions rely on an approximate filtering posterior rather than the exact smoothing distribution, which may be less accurate when future observations would substantially inform latent state estimates.

### G.2. Inhomogeneous forecasting distribution

In its presented form, ARPLN-ICA assumes a time-homogeneous switching prior over the latent regimes, which is mispecified for forecasting settings where an exogenous perturbation occurs at a fixed time, as in the gnotobiotic mice dataset. To mitigate this mismatch post-training, we replace the homogeneous regime prior with the empirical time-inhomogeneous Markov model of Eq. (61) obtained by aggregating filtering variational posteriors over the training trajectories, effectively incorporating time-inhomogeneous transitions in an online framework. The derivation is given in Appendix F.2 and yields an explicit parameterization which does not require prior knowledge on perturbation times. Combining the filtering variational approximation from Section 3.4 with the proposed post-hoc approximation for inhomogeneous switching, the one-step-ahead forecasting distribution $f_1(x_{1:T}) = p_{\boldsymbol{\theta}}(x_{T+1} \mid x_{1:T})$ can be approximated by

$$f_1(x_{T+1}) \approx \int p_{\boldsymbol{\theta}}(x_{T+1} \mid s_{T+1}) \, p_{\boldsymbol{\theta}}(s_{T+1} \mid s_T, u_{T+1}) \, \bar{q}_{\boldsymbol{\varphi}}(u_{T+1} \mid u_T) \, q_{\boldsymbol{\varphi}}(s_T \mid x_{1:T}) \, q_{\boldsymbol{\varphi}}(u_T \mid x_{1:T}) \, \mathrm{d}s_{T:T+1} \mathrm{d}u_{T:T+1} \ . \tag{65}$$

While for a given $h > 0$, $f_h$ is not available in closed form, we compute a deterministic forecast together with time-resolved uncertainty by recursively propagating regime-conditioned moments and performing moment matching under the induced regime mixture derived in the following Appendix G.3.

### G.3. Deterministic forecast via moment-matched mixture

For a given observation $x_{1:T}$, label switching at the last observed times has distribution $\alpha_{T,k}(x) = q_{\boldsymbol{\varphi}}(u_T = k \mid x_{1:T})$. Up to the horizon $h > 0$, we can compute the distribution of the switching states from the empirical inhomogeneous switching approximation as

$$\hat{\alpha}_{T+1}(x) = \bar{\tau}_{T+1} \, \alpha_T(x) \ , \quad \hat{\alpha}_{T+h}(x) = \bar{\tau}_{T+h} \, \hat{\alpha}_{T+h-1}(x) \ . \tag{66}$$

Then, to produce a mean deterministic forecast of the latent states $(s_t)_{T+h \geq t \geq T}$, we consider the mean trajectory given by $\hat{\mu}_{T+1:T+h}(x) = \mathbb{E}[s_{T+1:T+h} \mid x_{1:T}]$, which is recursively computed using the tower property. For the $i$-th component,

using the Markov structure of $(s_t, u_t)_{t \leq T+h}$, the variational proxy and the empirical inhomogeneous approximation yields

$$
\begin{aligned}
\hat{\mu}_{T+1}^{(i)}(x) &= \mathbb{E}\left[ \mathbb{E}\left[ s_{T+1}^{(i)} \mid x_{1:T}, u_{T+1}^{(i)}, s_T^{(i)}, u_T^{(i)} \right] \right] \\
&\approx \int \left( \mathbb{E}_{p_{\theta}}\left[ s_{T+1}^{(i)} \mid s_T^{(i)}, u_{T+1}^{(i)} \right] q_{\varphi}(s_T^{(i)} \mid x_{1:T}) \mathrm{d}s_T \right) \bar{q}_{\varphi}(u_{T+1}^{(i)} \mid u_T^{(i)}) \, q_{\varphi}(u_T^{(i)} \mid x_{1:T}) \mathrm{d}u_{T:T+1}^{(i)} \\
&= \sum_{k=1}^{C} \hat{\alpha}_{T+1,k}^{(i)}(x_{1:T}) \left( B_k^{(i)} \mu_T^{(i)}(x_{1:T}) + b_k^{(i)} \right) ,
\end{aligned}
\tag{67}
$$

where $\mu_T^{(i)}(x) = \mathbb{E}_{q_{\varphi}}[s_T \mid x_{1:T}]$. Using the same arguments, we derive a recursive formula to build the mean trajectory at step $t \geq 1$ given by

$$
\hat{\mu}_{T+t}^{(i)}(x) = \sum_{k=1}^{C} \hat{\alpha}_{T+t,k}^{(i)}(x_{1:T}) \left( B_k^{(i)} \hat{\mu}_{T+t-1}^{(i)}(x_{1:T}) + b_k^{(i)} \right) .
\tag{68}
$$

Similarly, we derive the variance of the forecasted latent trajectory at time $t \geq 1$ denoted $\hat{\psi}_{T+t}(x) = \mathbb{V}\mathrm{ar}[s_{T+t} \mid x_{1:T}]$,

$$
\hat{\psi}_{T+t}^{(i)}(x) \approx \sum_{k=1}^{C} \left[ \hat{\alpha}_{T+t,k}^{(i)}(x) \left( \psi_k^{(i)} + (B_k^{(i)})^2 \hat{\psi}_{T+t-1}^{(i)}(x) + \left( B_k \hat{\mu}_{T+t-1}^{(i)}(x) + b_k^{(i)} \right)^2 \right) \right] - (\hat{\mu}_{T+t}^{(i)})^2 ,
\tag{69}
$$

where $\hat{\psi}_{T+1}^{(i)}(x) = \sum_{k=1}^{C} \left[ \hat{\alpha}_{T+1,k}^{(i)}(x) \left( \psi_k^{(i)} + (B_k^{(i)})^2 \tilde{\psi}_T^{(i)}(x) + \left( B_k \mu_T^{(i)}(x) + b_k^{(i)} \right)^2 \right) \right] - (\hat{\mu}_{T+1}^{(i)})^2$.

To prevent excessive smoothing across regimes, we apply Viterbi algorithm to the Markov distribution $\hat{p}(u_{1:T+h} \mid x_{1:T})$, initialized by the marginal $\alpha_1(x)$ and driven by the time-dependent transition kernel

$$
\hat{\tau}_t = \tau_t(x)\, \mathbb{1}_{t \leq t_0} + \bar{\tau}_t\, \mathbb{1}_{t > t_0}.
$$

This yields the maximum a posteriori switching-state sequence, which can be used in place of the soft assignments $\hat{\alpha}_t$. As a result, the mean and variance of the predicted trajectory can exhibit sharp regime changes rather than gradual interpolation.

### G.4. Mitigating offset noise in microbiome analysis

In microbiome sequencing, the total read count varies across samples and time due to sampling effort and technical factors, and is not a direct proxy for absolute microbial count. Sequencing depth is therefore generally treated as a noise effect captured by a known exogenous offset. For each time point $t$, we consider the estimated offset

$$
o_t = \log \left( \sum_{k=1}^{K} x_t^{(k)} \right) ,
$$

which is assumed observed at test time and used when mapping predicted compositions back to counts. To mitigate depth-induced variability during training and enable fair comparison across baselines, CLR-VAR(1), UWEDGEICA and PICARD are trained in the Aitchison geometry using the CLR transform (see Eq.(74)). At prediction time, CLR forecasts are mapped to proportions via the softmax transform (invert of CLR) and then converted to count space by rescaling with the observed depth $\exp(o_t)$. Similarly, gLV is trained on depth-normalized counts, and its forecasts are rescaled using the same oracle test offset. Finally, ARPLN-ICA and P-rSLDS are trained with the explicit additive logsum offset effect. This procedure ensures comparability across methods while mitigating sensitivity to depth variation.

### G.5. Training protocol and architecture design

For the forecasting experiments, we use the filtering variational distribution introduced in Appendix G.1. The architecture follows Figure 10, with the key modification that the sequential encoder treats only past observations: at time $t$, it processes only the observed counts $x_{1:t}$ to parameterize the auto-regressive variational parameters.

During training, we adopt the same architectural choices as in the reconstruction experiments: a 3-layer GRU produces a sequence embedding of dimension 10, followed by a linear layer prior to concatenation with the current observation.

The recognition network is a 2-layer ReLU network with hidden sizes $[32, 16]$, and each head network is a 2-layer ReLU network with hidden width 16. Following Appendix G.4, ARPLN-ICA is fitted with logsum offsets, and fixed effects to model baseline abundance (see Section 3.4). We train for 800 epochs using AdamW with learning rate $10^{-3}$, weight decay at $10^{-4}$, gradient clipping at 5, and cosine annealing with $T_{\max} = 720$.

Similarly to Appendix F.4, we perform a quantitative benchmark to assess the effect of the number of latent sources on forecasting performance. For varying values of $d$, Figure 19 reports the trajectory-level metrics from Appendix H, comparing the forecasted trajectory from $t_0 = 2$ to the ground truth. Performance varies across metrics, but $d = 7$ appears like a good compromise between dimension reduction and forecasting efficacy.

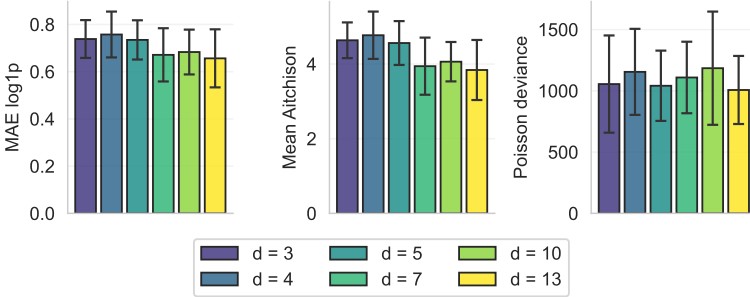

*Figure 19.* Average forecasting performance on held-out mice in the LOO-CV when forecasting from $t_0 = 2$, as a function of the latent dimension $d$. Bars report the mean and standard deviation across LOO splits of the metric value averaged over the test trajectory. All models share the same architecture and training protocol.

Figure 20 shows a representative forecasts for three microbial taxa from a held-out mouse, initialized at prediction times $t_0 = 2$ and $t'_0 = 13$. In both cases, ARPLN-ICA generally predicts the main abundance shifts associated with the perturbation, while showing relatable trends with the true trajectory.

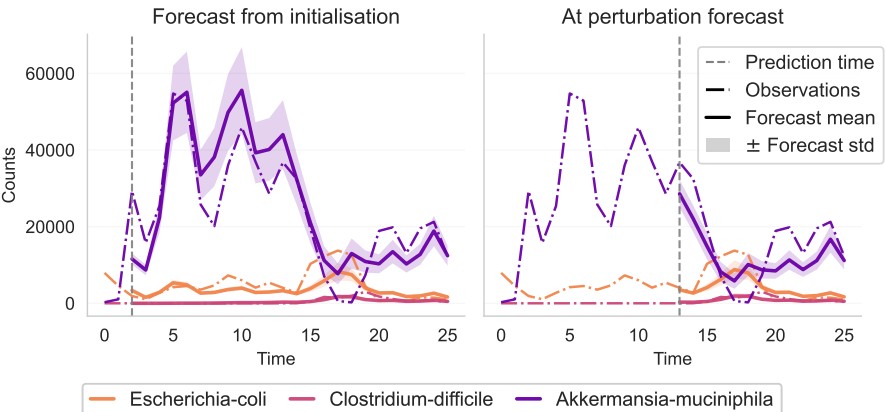

*Figure 20.* Illustration of forecasting tasks with *E. Coli*, *C. difficile* (inflammatory) and *C. scindens*, *R. hominis* (regulatory) from one gnotobiotic mouse. Dash–dot curves show observed test counts and solid curves show the deterministic forecast mean from ARPLN-ICA with standard deviation.

### G.6. Forecasting baselines for microbiome data

During forecasting, we benchmark ARPLN-ICA against a set of standard microbiome forecasting baselines. The selected methods span common modeling approaches used in practice, including auto-regressive models, ICA baselines, mechanistic approaches, as well as an ablation of regime switching in ARPLN-ICA.

**Time-homogeneous ARPLN-ICA.** As a first competitor, we consider a time-homogeneous variant of ARPLN-ICA obtained by disabling regime switching ($C = 1$). All other aspects of the model and training protocol are kept identical

to the reference ARPLN-ICA setting. This ablation isolates the contribution of regime inference and assesses whether switching dynamics improve forecasting performance.

**Time-varying CLR-VAR(1).**   As a representative auto-regressive baseline, we consider a time-varying vector auto-regressive model of order one (VAR(1)). For real-valued observations $Y = (Y_t)_{1 \leq t \leq T} \in \mathbb{R}^{T \times K}$, a time-inhomogeneous VAR(1) defines a Gaussian Markov chain

$$Y_1 \sim \mathcal{N}(m, \psi_1) \ , \quad Y_{t+1} \mid Y_t \sim \mathcal{N}(A_{t+1} Y_t + b_{t+1}, \psi_{t+1}) \ , \tag{70}$$

where $A_{t+1}, \psi_{t+1} \in \mathbb{R}^{K \times K}$ and $b_{t+1} \in \mathbb{R}^K$. A standard estimation strategy for this inhomogeneous specification is to fit $(A_{t+1}, b_{t+1})$ via ridge-penalized least squares (regularization at $10^{-3}$) and estimate $\psi_{t+1}$ by maximum likelihood (Haslbeck et al., 2021). Directly applying VAR models to raw counts is typically ill-suited due to positivity constraints, and heavy-tailed nature of count observations. Therefore, we rely on the CLR-VAR(1) (see Zheng & Chen (2017)): we fit VAR(1) in the Aitchison geometry (Aitchison, 1982) by first mapping counts to the simplex via the centered log-ratio (CLR) transform (Eq. (74)), which yields a Euclidean representation and mitigates compositional effects. Forecasts are then mapped back to the count space by applying the softmax transform (inverse of CLR) to obtain a composition, and scaling it by the exponential of the observed offset at each time $t$. The resulting method is denoted CLR-VAR(1) in our experiments.

**Linear ICA models in CLR space.**   Standard linear ICA tools can be used to analyze temporal data, notably UWEDGEICA (Pfister et al., 2019) and PICARD (Ablin et al., 2018) (see Appendix E.2). Although they do not define generative models by themselves, they can be used as a preprocessing step to obtain a source representation on top of a forecasting model such as the time-varying CLR-VAR(1) baseline. First, we map training counts to the CLR space. Then, we apply the ICA method to learn a linear demixing transform, encode the CLR trajectories into estimated sources, and fit a time-varying VAR(1) model on the training sources representation. For forecasting, we simulate source trajectories from the fitted VAR(1), decode them back to the CLR space using the inverse ICA transform, and finally map CLR forecasts to counts via the softmax transform followed by rescaling with the exponential offset at each time step. We refer to this pipeline as CLR-UWEDGEICA and CLR-PICARD in our experiments. Fitting steps are the same as that of Appendix E.2, with the additional CLR instead of log preprocessing. The baselines are implemented from the COROICA package (Pfister et al., 2019) and `python-picard` package (Ablin et al., 2018).

**Poisson rSLDS.**   Several models have been proposed for count data forecasting (see the review of Davis et al. (2021)). In particular, Poisson mixed models combined with switching latent dynamical systems have been successfully applied in biological settings, including neural spike modeling (Zoltowski et al., 2020; Glaser et al., 2020). Since ARPLN-ICA can be viewed as a Poisson log-link auto-regressive model with ICA-compatible switching, we consider an auto-regressive Poisson GLM model using a log-link combined with recurrent switching linear dynamical systems (P-rSLDS, Linderman et al. (2017)). This model uses grouped switching, which is not compatible with an ICA setting, but provides a more flexible model for count predictions. We implement this baseline using the SSM package, available at `github.com/lindermanlab/ssm`. In particular, offsets are incorporated as additive log-size effects on the log-intensities, as in ARPLN-ICA. The model is trained using the recommended hyperparameters of SSM, with the number of discrete states set equal to that of ARPLN-ICA to ensure comparability.

**Generalized Lotka-Volterra.**   Generalized Lotka-Volterra (gLV) models are widely used to describe ecological dynamics and have been extensively applied in microbiome analysis (Stein et al., 2013; Gibson et al., 2025). Here, we adopt the ridge-regularized gLV approach of Stein et al. (2013), which models the dynamics of microbial abundances as

$$\mathrm{d}x_t = x_t \odot (r + A x_t) \, \mathrm{d}t + \mathrm{d}B_t \ , \tag{71}$$

where $r \in \mathbb{R}^K$ denotes species-specific growth rates, $A \in \mathbb{R}^{K \times K}$ models pairwise interaction coefficients, and $B_t$ is a $K$-dimensional Brownian motion. Parameter estimation is performed with the ridge-penalized procedure of Stein et al. (2013), using the implementation provided in the MIMIC Python library (Fontanarrosa et al., 2025). In practice, we fit the model on normalized proportion data and map predictions back to the count scale by multiplying by the exponential offset, effectively mitigating sequencing-depth effects. We refer to this baseline as gLV-L2.

## H. Quantitative comparison of temporal count data

Given a reference temporal count trajectory $x = x_{1:T}^{1:K}$, we aim to quantify how close a reconstructed or forecasted trajectory $\hat{x} = \hat{x}_{1:T}^{1:K}$ is to $x$. In this context, we report complementary metrics that capture (i) point-wise discrepancies along time, (ii) goodness-of-fit under a Poisson reference, and (iii) community-level dissimilarity relevant to microbiome analysis.

**Mean Absolute Error.** The mean absolute error (MAE) is a standard regression metric that summarizes point-wise deviations across time and tracked entities given by

$$\text{MAE}(x, \hat{x}) = \frac{1}{TK} \sum_{t=1}^{T} \sum_{k=1}^{K} \left| x_t^{(k)} - \hat{x}_t^{(k)} \right|. \tag{72}$$

Notably, to account for the overdispersion effect of count data and mitigate offset noise, we compute the MAE on $\log(\cdot + 1)$ counts.

**Poisson deviance.** Due to their discrete nature, count data are typically modeled using Poisson distribution. To evaluate fit under this reference model, we treat $\hat{x}_t^{(k)}$ as the fitted mean for $x_t^{(k)}$ and compute the average Poisson deviance given by

$$\text{D}_{\mathcal{P}}(x, \hat{x}) = \frac{2}{TK} \sum_{t=1}^{T} \sum_{k=1}^{K} \left( x_t^{(k)} \log \left( x_t^{(k)} / \hat{x}_t^{(k)} \right) - x_t^{(k)} + \hat{x}_t^{(k)} \right), \tag{73}$$

with the convention that the contribution is set to 0 whenever $x_t^{(k)} = 0$ and $\hat{x}_t^{(k)} > 0$. This metric aggregates discrepancies between observed counts and their Poisson fitted means across time and tracked entities.

**Aitchison distance.** In microbiome studies, community-level dissimilarity is often assessed through $\beta$-diversity measures (Bastiaanssen et al., 2023). In this work, we focus on the Aitchison distance, which is a mathematically valid distance on the simplex defined via the centered log-ratio (CLR) transform. For temporal trajectories, we report the mean Aitchison distance given by

$$\text{D}_{\text{CLR}}(x, \hat{x}) = \frac{1}{T} \sum_{t=1}^{T} \| \text{clr}(x_t) - \text{clr}(\hat{x}_t) \|_2, \tag{74}$$

where $\text{clr}(x_t) = \left[ \log x_t^{(k)} - \frac{1}{K} \sum_{j=1}^{K} \log x_t^{(j)} \right]_{1 \le k \le K}$. In practice, we apply the CLR transform after adding a small pseudo-count of 0.5 to handle zeros.

## I. Climate science dataset experiments

### I.1. Storm Events dataset description

The NOAA Storm Events database records severe weather hazards occurring across the United States since 1950, counting major, unusual and rare weather hazards (NOAA et al., 1996). In this work, we consider the subset of yearly panels from 2000 to 2025 ($n = 25$) for $K = 16$ continental event types, aggregated at the monthly scale ($T = 12$). Table 1 reports the corresponding monthly means and standard deviations, showing highly heterogeneous profiles across event types, months and years. Notably, several event types exhibit many low or near-zero monthly counts, reflecting the sparsity induced by rare or strongly seasonal phenomena. These characteristics make the dataset well suited to ARPLN-ICA, which can model heterogeneous count profiles while extracting recurrent seasonal regimes and co-occurrence patterns. Such dependencies may reflect shared meteorological drivers, regional effects, or longer-term changes in climate conditions.

*Table 1.* Monthly mean event counts with standard deviations for the NOAA Storm Events dataset between 2000 and 2025.

| Weather hazard | January | February | March | April | May | June | July | August | September | October | November | December |
|---|---|---|---|---|---|---|---|---|---|---|---|---|
| Cold/Wind Chill | 246.0 (260.8) | 111.6 (164.3) | 14.5 (19.6) | 9.0 (25.2) | 7.3 (21.7) | 2.6 (8.6) | 5.2 (17.8) | 2.0 (9.2) | 0.4 (1.3) | 9.4 (17.2) | 4.7 (9.9) | 77.5 (104.8) |
| Drought | 202.3 (173.5) | 189.5 (153.2) | 196.3 (150.8) | 204.7 (149.8) | 190.4 (145.4) | 197.7 (164.3) | 246.7 (234.2) | 304.9 (232.7) | 317.0 (224.7) | 319.2 (226.3) | 297.0 (243.5) | 249.2 (198.6) |
| Excessive Heat | 7.5 (26.2) | 6.1 (22.7) | 4.6 (23.5) | 4.1 (12.3) | 12.2 (28.3) | 142.0 (192.0) | 323.7 (387.9) | 275.3 (576.9) | 36.2 (53.1) | 2.7 (7.3) | 0.1 (0.4) | 0.1 (0.3) |
| Flash Flood | 92.4 (79.8) | 106.7 (85.1) | 160.5 (111.7) | 254.7 (167.0) | 498.5 (220.6) | 601.2 (199.4) | 717.6 (282.6) | 599.0 (203.0) | 390.6 (160.3) | 171.5 (135.6) | 82.5 (69.7) | 87.6 (91.6) |
| Flood | 165.9 (140.1) | 213.0 (195.0) | 268.9 (185.8) | 259.6 (140.5) | 314.0 (158.7) | 279.5 (140.0) | 182.8 (86.1) | 158.7 (90.0) | 184.6 (173.2) | 116.9 (90.1) | 85.9 (57.9) | 153.2 (104.8) |
| Hail | 65.4 (77.0) | 180.6 (149.4) | 861.9 (450.0) | 1797.9 (762.9) | 2700.2 (788.5) | 2389.5 (1065.8) | 1495.3 (492.4) | 1051.5 (395.4) | 462.9 (202.1) | 213.0 (126.1) | 95.3 (74.0) | 60.1 (64.4) |
| Heavy Rain | 31.8 (18.8) | 40.4 (26.4) | 61.0 (44.3) | 53.3 (34.3) | 102.7 (46.2) | 143.5 (65.6) | 183.8 (100.1) | 172.6 (91.3) | 138.1 (112.3) | 76.5 (55.4) | 50.1 (39.2) | 49.6 (45.7) |
| Heavy Snow | 591.7 (223.9) | 569.5 (255.1) | 349.5 (160.7) | 114.6 (63.8) | 27.4 (21.6) | 4.4 (10.2) | 0.0 (0.0) | 0.2 (0.5) | 6.0 (8.8) | 63.9 (55.8) | 194.1 (91.3) | 510.5 (264.4) |
| High Wind | 411.2 (223.6) | 435.3 (213.3) | 459.0 (323.3) | 405.6 (279.9) | 161.5 (130.4) | 93.6 (78.9) | 19.7 (16.3) | 25.0 (23.8) | 94.5 (88.6) | 281.8 (155.8) | 350.7 (178.4) | 507.0 (286.1) |
| Thunderstorm Wind | 246.5 (244.8) | 316.4 (248.7) | 645.6 (539.5) | 1268.6 (762.9) | 2212.0 (865.8) | 3672.0 (1000.7) | 3521.6 (1017.7) | 2227.8 (726.9) | 692.7 (337.5) | 359.5 (254.3) | 260.1 (198.3) | 219.6 (215.0) |
| Tornado | 44.3 (48.8) | 48.4 (41.7) | 121.9 (88.1) | 241.8 (180.0) | 313.4 (162.6) | 201.0 (142.4) | 114.8 (42.3) | 91.1 (44.0) | 69.1 (60.9) | 66.9 (41.4) | 64.8 (50.9) | 57.5 (67.9) |
| Winter Storm | 759.3 (323.9) | 788.1 (409.6) | 411.5 (234.2) | 181.5 (135.4) | 26.1 (26.4) | 0.7 (1.4) | 0.0 (0.0) | 0.0 (0.2) | 6.9 (19.1) | 56.3 (55.3) | 188.2 (112.0) | 621.0 (352.3) |
| Blizzard | 100.5 (87.8) | 110.0 (103.6) | 84.5 (87.0) | 39.2 (44.2) | 1.9 (4.3) | 0.0 (0.0) | 0.2 (0.8) | 0.0 (0.0) | 0.2 (0.8) | 9.5 (14.9) | 35.5 (43.1) | 158.2 (159.0) |
| Dense Fog | 110.9 (52.5) | 65.9 (48.3) | 50.8 (39.9) | 27.0 (28.1) | 27.0 (28.1) | 15.8 (20.7) | 27.3 (25.7) | 32.0 (35.1) | 44.5 (39.4) | 74.5 (63.9) | 129.1 (88.6) |
| Frost/Freeze | 41.8 (30.9) | 25.2 (24.0) | 24.6 (30.0) | 196.2 (299.8) | 58.6 (67.9) | 10.7 (13.3) | 0.0 (0.0) | 1.1 (3.7) | 15.2 (14.0) | 123.8 (62.5) | 45.9 (34.9) | 28.4 (23.1) |
| Lightning | 5.0 (4.8) | 7.3 (5.8) | 16.1 (9.1) | 33.5 (17.3) | 64.3 (37.6) | 114.1 (63.9) | 148.5 (63.5) | 118.2 (63.0) | 35.7 (18.8) | 12.4 (8.0) | 4.5 (4.2) | 3.8 (3.0) |

## I.2. Model selection and ICA stability

As in Appendix F.4 for microbiome data analyses, the choice of the latent dimension $d$ is an important design parameter in ICA. Since there is no universal selection criterion, we choose $d$ using reconstruction metrics relevant to the application. For the Storm Events dataset, we consider only MAE and Poisson deviance computed on test samples (see Appendix H), as Aitchison distance is specific to compositional data and is therefore not applicable with climate events. The choice $d = 4$ provides a reasonable trade-off between dimensionality reduction, interpretability, and reconstruction quality, while $d = 10$ yields better reconstruction in this setting at the cost of more involved interpretability.

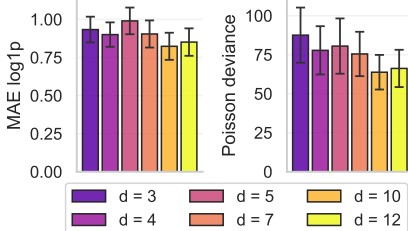

*Figure 21.* Average reconstruction performance on test year from LOO CV on the Storm Events dataset, with increasing dimensionality of the latent sources. Barplots show the mean and standard deviation across LOO splits of the scoring value averaged along the test year trajectory. All methods are evaluated with the same parameterization, with varying $d$.

We then assess the stability of ARPLN-ICA on the Storm Events dataset using leave-one-out cross-validation (LOO-CV) at $d = 4, C = 2$, yielding 25 estimated mixing matrices $\Gamma$, one for each split (same hyperparameters as Appendix 10). Figure 22 reports pairwise absolute cosine similarities between the estimated mixings, averaged across columns, together with column-wise average similarities and 95% confidence intervals. The recovered mixings are highly stable across folds, with a mean similarity above 99.7%, supporting the fold-wise comparability of the inferred components.

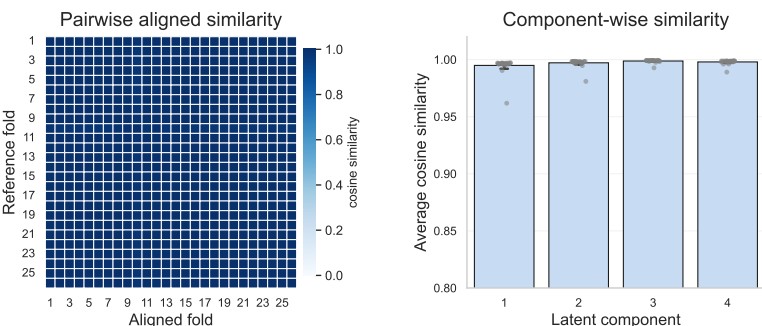

*Figure 22.* Stability of the learned mixing matrix $\Gamma$ across leave-one-out folds on the Storm Events dataset at $d = 4$. (Left) Pairwise cosine similarity between $\Gamma$ estimates after aligning fold $j$ to fold $i$ (permutation/sign invariances). (Right) Component-wise stability, computed as the mean absolute cosine similarity of each aligned column, aggregated within each reference fold and summarized across folds, with 95% confidence interval and similarity per fold indicated by grey dots.

