# OpenReview forum: "Independent Component Discovery in Temporal Count Data"
_ICML.cc/2026/Conference — ICML 2026 regular_

### Official Review · Reviewer_usAC · 2026-03-07

**Soundness:** 3
**Presentation:** 3
**Significance:** 3
**Originality:** 3
**Overall Recommendation:** 4
**Confidence:** 3

**Summary:**

This paper studies the inference of the underlying dynamical system based on temporal count data. The link function is a Poisson GLM and latent dynamics are linear or switching linear. The dimensionality of the latent representation and the number of discrete states in the switching mode are hyperparameters of the model. The paper demonstrates its findings on a synthetic dataset and a dataset of mouse gut microbiome (Bucci et al, 2016), classified into 13 different bacterial taxa via genome sequencing, observed at 26 time points. The Bucci dataset introduces an infection halfway through the experiment, so the authors consider 2 states in some of their analyses.

**Compliance With Llm Reviewing Policy:**

Affirmed.

**Final Justification:**

The rebuttal has addressed my main concerns. I have accordingly updated my score.

**Key Questions For Authors:**

- Could the authors compare against a Gaussian observation model? (e.g., count data can be centered and normalized) Does the present model offer quantitative or qualitative gains over this baseline for the microbiome dataset?

- Could the authors compare against existing Poisson GLM and recent switching LDS models (rslds, p-dlds)?

**Limitations:**

yes

**Strengths And Weaknesses:**

- Temporal count data is frequently encountered in neurobiology where the spiking activity of individual neurons is commonly represented as counts per time bins. There is a rich literature on the analysis of such datasets, including the Poisson GLM model used here. This literature is completely missing here. (e.g., Pillow, Paninski, Linderman groups, among others)

- [EDIT: The authors have successfully addressed this point.] Related to the comment above, multiple highly relevant models are ignored here. (e.g., rslds, p-dlds, gp-rslds, among others) These would constitute strong baselines. I believe the present study mostly recapitulates those earlier findings, which significantly decreases its originality and significance.

- [EDIT: The authors have conducted a new experiment to address this point.] The motivation of the count model is not well established. For the main and the only real data application of the manuscript, counts are in the multiple 10000s (Figure 19). Since the Poisson approaches the Gaussian at such high counts, I wonder what one gains by not using a simpler Gaussian observation model.

- I believe the theoretical treatment is already common knowledge in the Poisson GLM literature. I appreciate the proofs, although I have one minor comment on presentation: these are simple observations/proofs. I am not sure if dividing them into multiple lemmas, each needing a few lines to prove, improves the presentation.

---

> ### Author Rebuttal · Authors · 2026-03-30
>
> Dear reviewer, thank you for your review.
>
> We believe that the comment **"the theoretical treatment is already common knowledge in the Poisson GLM literature"** suggests that our contribution may not have been fully understood. Our model builds on recent advances in structured ICA, making it far from generic. The recent literature and the comments from other reviewers highlights that extending ICA to complex settings such as temporal count data is challenging both for practical and theoretical reasons. We agree that there exist neighboring models combining Poisson log-link with SLDS, and that additional references and benchmarks would improve the positioning of our work. Yet unifying these approaches for ICA, developing identifiability guarantees in that setting, and learning with a structured variational inference framework, are novel contributions that are not present in the existing literature to the best of our knowledge. We hope that the proposed modifications addressing all comments will clarify this point. Please find our responses to your remarks below:
>
> **"There is a rich literature on the analysis of [temporal count data in neurobiology], including the Poisson GLM model used here. This literature is completely missing here."**
>
> The works you mention from the Pillow, Paninski, and Linderman groups are relevant neighboring literature on Poisson latent dynamical models, but they target different objectives. These methods were primarily introduced as flexible dynamical models for neural spike activity, whereas our framework is explicitly built around ICA factorization of the log-intensities into independent latent sources for downstream analysis. To the best of our knowledge, there is currently no ICA method for temporal count data with this objective.
>
> While we made an initial connection to Poisson GLM time-series models through the review of [3], we agree that the paper would benefit from a broader discussion of adjacent work on Poisson latent dynamical systems, and we will revise the related-work section accordingly to make our positioning clearer.
>
> **"these are simple observations/proofs ..."**
>
> While some intermediate lemmas are short, the identifiability contribution lies in how they connect to PLN-ICA identifiability. We separated the argument into short lemmas for clarity, though we agree parts of the proofs can be tightened.
>
> **"Could the authors compare against existing Poisson GLM and recent switching LDS models"**
>
> Thank you for pointing to these baselines. They are relevant neighboring SLDS models, although not ICA-compatible baselines, since ICA requires the factorization $p_\theta(s)=\prod_{i=1}^{d} p_\theta(s^{(i)}),$ which underlies source identification. Still, we agree they could be informative for the auxiliary forecasting evaluation. Therefore, we added an rSLDS baseline with Poisson log-link using the SSM library of Lindermanlab, matching the latent dimension and number of regimes of our model. Results are competitive although slightly less than ARPLN-ICA as reported in the table below.
>
> |Model|MAE log1p ↓|Mean Aitchison ↓|Poisson deviance ↓|
> |---|---:|---:|---:|
> |ARPLN-ICA|0.394 (0.102)|2.221 (0.609)|747.249 (297.830)|
> |(h)-ARPLN-ICA|0.410 (0.103)|2.334 (0.571)|869.655 (428.296)|
> |**rSLDS**|0.411 (0.104)|2.188 (0.614)|1075.468 (407.643)|
> |CLR-VAR(1)|0.385 (0.032)|1.810 (0.043)|1640.757 (647.301)|
> |CLR-UwedgeICA|0.486 (0.128)|2.082 (0.222)|2658.846 (1565.574)|
> |CLR-Picard|0.547 (0.137)|2.240 (0.250)|3570.374 (1980.226)|
> |gLV-L2|0.487 (0.083)|2.553 (0.474)|1847.153 (242.176)|
>
> **Table 5:** Forecasting results as mean (std) over LOO.
>
> **"The motivation of the count model is not well established. [...] Since the Poisson approaches the Gaussian at such high counts [...] Could the authors compare against a Gaussian observation model?"**
>
> While a Poisson distribution with large mean can be approximated by a Gaussian at first order, this does not imply that a Gaussian observation model is equally appropriate for this dataset. Many taxa still reach low abundances where this asymptotic approximation is poor (see Figures 15 and 19), and the data remain discrete, non-negative, overdispersed, and strongly heteroskedastic across taxa and time, which is not modeled by Gaussian distributions.
>
> Importantly, the comparison requested in the review is already partly addressed in the manuscript through Gaussian baselines in normalized Aitchison space. The need for count-data models is also supported empirically in the synthetic experiments, where methods based on linear ICA underperform.
>
> We agree that this comparison could be made more explicit in the paper. In the revision, we will clarify that CLR-VAR(1) serves as a Gaussian baseline in the auxiliary forecasting task, and will improve the discussion of what is gained by using a count-specific observation model on this dataset.
>
> [3] : Davis et al. "Count time series: A methodological review". Journal of the American Statistical Association (2021)

---

> > ### Author Rebuttal · Reviewer_usAC · 2026-04-01
> >
> > Thank you for the responses. These have indeed addressed my confusion regarding switching models. Accordingly, I will increase my score to 3.
> >
> > I still think the application/chosen dataset is not well suited: ideally, I'd like to see an application with much lower counts (e.g., <10 per bin). The broader point is that I find it hard to tell if the manuscript adds a truly useful method. I would much prefer to see either results based on much lower counts, or a comparative study with a Gaussian model on this dataset where the Gaussian model fails to uncover the same qualitative results.

---

> > > ### Author Response · Authors · 2026-04-03
> > >
> > > Dear reviewer, thank you for re-evaluating our work.
> > >
> > > To address your concern that the microbiome application may not isolate the advantage of a count model clearly enough, we note that a second real-data application was added during the rebuttal based on the NOAA Storm Events dataset monitoring 16 continental weather hazards monthly over 25 years. This dataset exhibits strong seasonal heterogeneity, with several event types having low or near-zero counts during part of the year (see Table 6 below), while others enter high-activity regimes, making it a natural setting for count-based modeling.
> > >
> > > Qualitatively, the model recovers meaningful seasonal structure and plausible cross-hazard relationships (see analyses in rebuttals qTS7 and LkU2). Quantitatively, following the same protocol as in the manuscript, we compared ARPLN-ICA against Gaussian VAR(1) + ICA baselines on normalized counts.  The results below show improved predictive fit for ARPLN-ICA over Gaussian baselines on this dataset.
> > >
> > > Together, these results support the practical interest of the method in another real temporal count setting with clear sparsity patterns and low counts.
> > >
> > > We hope this addresses your concern.
> > >
> > > |Hazard|Jan|Feb|Mar|Apr|May|Jun|Jul|Aug|Sep|Oct|Nov|Dec|
> > > |---|---:|---:|---:|---:|---:|---:|---:|---:|---:|---:|---:|---:|
> > > |Cold/Wind Chill|246.0|111.6|14.5|9.0|7.3|2.6|5.2|2.0|0.4|9.4|4.7|77.5|
> > > |Drought|202.3|189.5|196.3|204.7|190.4|197.7|246.7|304.9|317.0|319.2|297.0|249.2|
> > > |Excessive Heat|7.5|6.1|4.6|4.1|12.2|142.0|323.7|275.3|36.2|2.7|0.1|0.1|
> > > |Flash Flood|92.4|106.7|160.5|254.7|498.5|601.2|717.6|599.0|390.6|171.5|82.5|87.6|
> > > |Flood|165.9|213.0|268.9|259.6|314.0|279.5|182.8|158.7|184.6|116.9|85.9|153.2|
> > > |Hail|65.4|180.6|861.9|1797.9|2700.2|2389.5|1495.3|1051.5|462.9|213.0|95.3|60.1|
> > > |Heavy Rain|31.8|40.4|61.0|53.3|102.7|143.5|183.8|172.6|138.1|76.5|50.1|49.6|
> > > |Heavy Snow|591.7|569.5|349.5|114.6|27.4|4.4|0.0|0.2|6.0|63.9|194.1|510.5|
> > > |High Wind|411.2|435.3|459.0|405.6|161.5|93.6|19.7|25.0|94.5|281.8|350.7|507.0|
> > > |Thunderstorm Wind|246.5|316.4|645.6|1268.6|2212.0|3672.0|3521.6|2227.8|692.7|359.5|260.1|219.6|
> > > |Tornado|44.3|48.4|121.9|241.8|313.4|201.0|114.8|91.1|69.1|66.9|64.8|57.5|
> > > |Winter Storm|759.3|788.1|411.5|181.5|26.1|0.7|0.0|0.0|6.9|56.3|188.2|621.0|
> > > |Blizzard|100.5|110.0|84.5|39.2|1.9|0.0|0.2|0.0|0.2|9.5|35.5|158.2|
> > > |Dense Fog|110.9|65.9|50.8|21.5|27.0|9.4|15.8|27.3|32.0|44.5|74.5|129.1|
> > > |Frost/Freeze|41.8|25.2|24.6|196.2|58.6|10.7|0.0|1.1|15.2|123.8|45.9|28.4|
> > > |Lightning|5.0|7.3|16.1|33.5|64.3|114.1|148.5|118.2|35.7|12.4|4.5|3.8|
> > > **Table 6**: Average hazard occurrence per month over 25 years
> > >
> > > | Method | MAE log1p | Poisson deviance |
> > > |---|---:|---:|
> > > | ARPLN-ICA | 0.025 (0.003) | 2.13 (0.63) |
> > > | VAR(1) + UwedgeICA | 0.029 (0.005) | 3.77 (4.48) |
> > > | VAR(1) + Picard | 0.028 (0.004) | 3.44 (4.84) |
> > > **Table 7**: Multi-horizons forecasting performances aggregated per LOO CV (mean and std)

---

### Official Review · Reviewer_LkU2 · 2026-03-09

**Soundness:** 4
**Presentation:** 4
**Significance:** 3
**Originality:** 3
**Overall Recommendation:** 5
**Confidence:** 4

**Summary:**

The authors propose a new Independent Component Analysis (ICA) model for temporal count data based on the Poison-lognormal framework. In addition, the model can incorporate regime shift with discrete latent variables. Identifiability results are provided, and a variational approach for training is proposed. Finally empirical evaluation on simulated data and on a real life biological data is carried out.

**Compliance With Llm Reviewing Policy:**

Affirmed.

**Final Justification:**

The presented method is sound, sufficiently novel and the problem addressed is relevant. The authors answered my questions sufficiently well.
In accordance with my original assessment, I support acceptance.

**Key Questions For Authors:**

- Would be valuable to demonstrate the working of the method on another real life dataset given the small size of the biological one. You mentioned in the introduction that climate science or finance also uses this type of datasets. Can you execute a preliminary study on one of these? If the time is not sufficient it would be interesting to mention some publicly available dataset for further research.
- On Figure 3 the mentioned trends are not always clearly visible, in component 3 it is quite nice, but in component 1 and even in component 2 is nearly flat. This also makes me wonder if the model is even more powerful. It would be nice to see if the model still works in such small n as 2 or even with a single time series.
- Is there any evidence supporting a causal effect pointing from Clostridium dificile colonization towards reduced Akkermansia muciniphila numbers? The anti-inflamatory properties hypothesized by Derrien et al 2017 seems to suggest the other causal direction if it suggest anything.

**Limitations:**

yes

**Strengths And Weaknesses:**

Strength:
- The problem of analyzing count time series data is a relevant problem, and the tackled application metagenomics has its uses not only in healthcare, but in environmental sciences, agriculture and in many more areas.
- The presented method is sound, the modeling assumptions are clearly formulated.
- The work is well presented, easy to follow.

Weaknesses:
- Empirical evaluation is a bit weak, besides a simulation study, only a single relatively small experimental dataset is analyzed.
- The motivation given for forecasting is not convincing. This is not a serious issue, as it is just an auxiliary task here, but still worth to mention: using a model with quite restricted structure to generate data to use later is a risky suggestion.

---

> ### Author Rebuttal · Authors · 2026-03-30
>
> Dear reviewer, we very much appreciate your positive feedback on our work. Please find our answer to your questions below:
>
> **"Would be valuable to demonstrate the method on another real life dataset [...] Can you execute a preliminary study on one ?"**
>
> We thank the reviewer for this valuable suggestion. We agree that validating the method on a second real dataset would strengthen the practical relevance of our approach. We originally focused on the microbiome application because it lies within our domain expertise, which allowed us to complement the quantitative evaluation with biologically informed interpretation of the recovered components.
>
> To address this point, we conducted a preliminary analysis on the NOAA Storm Events database, a climate dataset tracking occurrences of severe weather events across the USA. We considered a subset of 16 continental event types aggregated monthly, then reorganized into annual trajectories to focus on seasonal structure. This yields a panel of yearly multivariate count series, on which we applied the same inference pipeline as in the microbiome study, following reviewer qTS7's transferability question, using 2 regimes, logsum offsets per year to mitigate climate change trends, and a Leave-One-Out cross-validation scheme. We explored several ICA dimensions and found that $d=4$ provides a good trade-off between interpretability and reconstruction quality.
>
> First, we looked into the ICA mixing. Due to space constraints, please refer to qTS7 rebuttal above for the analysis of the ICA mixing. We then examined regime categories in Table 3 below, showing a general warm-cold season contrast pattern for components 1, 2, and 4.
>
> Overall, this preliminary experiment provides evidence that ARPLN-ICA can extract meaningful latent structure from real multivariate count time series outside the biological domain. We will include a concise summary of this additional analysis in the revised manuscript, as well as possible other public datasets such as GDELT and Chicago crime records in social science to facilitate further research.
>
> |Comp.|Jan|Feb|Mar|Apr|May|Jun|Jul|Aug|Sep|Oct|Nov|Dec|
> |---|---:|---:|---:|---:|---:|---:|---:|---:|---:|---:|---:|---:|
> |C1|0.51|0.37|0.34|0.05|0.00|0.00|0.00|0.02|0.80|0.96|0.99|0.97|
> |C2|0.64|0.61|0.61|0.83|0.98|1.00|1.00|0.99|0.16|0.00|0.00|0.00|
> |C3|0.37|0.88|0.97|0.96|0.96|0.96|0.95|0.96|0.96|0.96|0.96|0.96|
> |C4|0.96|0.96|1.00|0.88|0.59|0.27|0.15|0.10|0.38|0.94|0.98|1.00|
>
> **Table 3:** Posterior switching probability of state 0 for each NOAA component.
>
> **"The motivation given for forecasting is not convincing ..."**
>
> We agree that the forecasting motivation should not be overinterpreted here. In our work, forecasting is included only as an auxiliary quantitative benchmark to assess whether the learned latent representation captures adequate temporal structure. Although we motivated it from the angle of data augmentation, it is not intended as a primary motivation for the method, and it may lack the biological faithfulness expected in clinical analysis, as you rightfully point out. We will clarify this point in the revised manuscript and state more explicitly the limits and goals of this evaluation protocol.
>
> **"On Figure 3 the trends are not always clearly visible [...] It would be nice to see if the model still works in small n"**
>
> We thank the reviewer for this suggestion. We agree that the trends are clearer for some components than for others. Following your suggestion, we conducted an additional experiment with a single training trajectory following our initial protocol (see concise table below). This additional limit case makes the overall trend clearer both in mean recovery and in stability across runs. We will revise the discussion of Figure 3 accordingly in the revised manuscript.
>
> ||C1|C2|C3|C4|C5|
> |---|---:|---:|---:|---:|---:|
> |n=1|0.73 (0.15)|0.48 (0.18)|0.64 (0.14)|0.41 (0.17)|0.87 (0.20)|
> |n=50|0.86 (0.06)|0.66 (0.07)|0.63 (0.15)|0.64 (0.08)|0.95 (0.09)|
> |n=150|0.86 (0.08)|0.65 (0.08)|0.67 (0.13)|0.65 (0.07)|0.97 (0.05)|
>
> **Table 4:** Average cosine similarity per component for varying $n$ (mean and std over 10 repetitions).
>
> **"Is there any evidence supporting a causal effect from C. difficile colonization towards reduced A. muciniphila"**
>
> We agree that biological interpretations should be stated carefully. To our knowledge, the current literature does not provide strong direct evidence for a directed causal effect from *C. difficile* colonization towards reduced *A. muciniphila* abundance.
>
> In our work, we discuss the corresponding entries of $\Gamma$ as taxa-level co-variations based on our identifiability results. Therefore, ARPLN-ICA can not support a directed causal claim here, and may only highlight a statistically meaningful association that is compatible with the microbiome literature and may motivate further biological investigation. We will ensure that this limitation is made clear throughout the revised manuscript.

---

> > ### Author Rebuttal · Reviewer_LkU2 · 2026-04-03
> >
> > Thank you for your detailed response.

---

### Official Review · Reviewer_sfAG · 2026-03-12

**Soundness:** 4
**Presentation:** 4
**Significance:** 3
**Originality:** 3
**Overall Recommendation:** 4
**Confidence:** 3

**Summary:**

This paper presents a novel framework for Independent Component Analysis (ICA) tailored to temporal count data, which are common in fields like microbiology, ecology, and healthcare. The authors propose a generative model that combines Poisson log-normal (PLN) emissions with linear mixing and regime-switching autoregressive dynamics on latent sources. The model, termed ARPLN-ICA, is designed to recover interpretable latent components while accounting for non-stationarity and overdispersion. Key contributions include: (i) a structured ICA model for count data with regime switching, (ii) theoretical identifiability results for the mixing matrix, and (iii) an amortized variational inference procedure with closed-form updates for most parameters. The method is evaluated on synthetic data and a real-world microbiome dataset, demonstrating improved recovery and interpretability over baseline ICA methods.

**Compliance With Llm Reviewing Policy:**

Affirmed.

**Key Questions For Authors:**

The runtime analysis in Appendix E.5 shows a sharp increase in per-epoch time beyond n≈300. While this is acceptable for small datasets, it may limit applicability to larger cohorts. The authors could comment on potential optimizations (e.g., mini-batching, sparse approximations) or future directions.

**Limitations:**

yes

**Strengths And Weaknesses:**

Strengths
1. Novelty of the Formulation: The ARPLN-ICA framework fills a well-documented gap by adapting ICA to temporal count data—an understudied setting where standard real-valued/additive noise ICA models fail. The integration of PLN emissions (to capture count statistics) and switching linear dynamics (to model regime shifts/perturbations) is a principled and innovative combination, with no direct prior work unifying these components for ICA.

2. Rigorous Theoretical Guarantees: The establishment of identifiability (Proposition 3.1, Corollary 3.2) is a major strength, as identifiability is a prerequisite for meaningful interpretability of latent components in ICA. The mild non-degeneracy assumptions for identifiability (e.g., positive diagonal initial covariance, distinct whitened lag-covariance diagonals) are generically satisfied, making the results practically relevant.

3. Efficient and Scalable Inference: The amortized variational inference procedure, which leverages GRU encoders for temporal feature extraction and coordinate ascent variational inference (CAVI) for discrete regime states, is well-designed for multivariate time series. The structured variational family (over mean-field) better captures temporal dependencies, and closed-form updates for most model parameters (excluding the mixing matrix) ensure computational efficiency.

Weaknesses
1. Failure to Break Through Linear Mixing Assumptions ：The paper uses a “linear mixing matrix Γ” to map latent sources to log-intensities, which simplifies identifiability proofs but restricts the model’s ability to capture complex non-linear dependencies. In temporal count data, relationships between latent components and observations may be non-linear (e.g., “threshold effects in microbial interactions”). The paper does not compare the performance of “linear mixing” with “non-linear mixing” (e.g., neural network-based mixing) nor clarify the applicable boundaries of the linear assumption, resulting in insufficient method flexibility.

2. Lack of Quantitative Validation for Biological Interpretation of Latent Components：
In the microbiome experiment, the paper claims that “Component 4 captures infection-driven mechanisms” but only provides qualitative interpretation through “microbial taxa co-variation patterns” and “alignment of state switches with infection timing.” It does not conduct quantitative validation with known biological knowledge: for example, is there a statistically significant association between the loadings of Component 4 and the functional annotations of microbial taxa (e.g., classification of pathogenic/beneficial bacteria)? Are component dynamics correlated with mouse physiological indicators (e.g., inflammatory factor levels)? Interpretations lacking quantitative validation may lead to subjective and accidental biological conclusions.

---

> ### Author Rebuttal · Authors · 2026-03-30
>
> Dear reviewer, we would like to thank you for this thorough evaluation of our work. We provide responses to your question and comments below:
>
> **"Failure to Break Through Linear Mixing Assumptions..."**
>
> We agree that non-linear mixing is a meaningful extension when the goal is to capture complex ICA dependencies in the latent log-intensity process. Our focus on the linear ICA setting here is primarily motivated by interpretability under mild assumptions, as you rightfully point out. However, we would like to emphasize that the sources follow a regime-switching mechanism, enabling globally non-linear dynamics. From a practical standpoint, linear ICA is also more robust in low-sample-size settings, as non-linear ICA models are typically more difficult to train reliably with limited data.
>
> Moreover, our work already provides theoretical foundations compatible with non-linear PLN-ICA extensions. As discussed in response to reviewer qTS7, Proposition 3.1 shows that identifiability in PLN-ICA reduces to identifiability of the noiseless latent log-intensity process. This result opens a path towards non-linear PLN-ICA models by combining our count observation model with existing identifiability results for noiseless non-linear ICA, such as Theorem 2 of [1]. We did not pursue this extension here because the goal of the present work is to establish a first identifiable and interpretable ICA framework for temporal count data under mild assumptions.
>
> Regarding comparisons with non-linear ICA methods, we would like to emphasize that our simulation study is designed to evaluate recovery of the mixing function under the model class considered in the paper. In that context, comparing against methods tailored to a different mixing family would not be apples-to-apples. While we could include a non-linear ICA baseline in the auxiliary forecasting experiment, architecture selection for neural-based mixing is difficult due to the small size of the dataset. Additionally, forecasting is not the primary objective of ICA, so such comparisons would remain secondary to the main methodological contribution.
>
> Following your remark, we will clarify the interest and limits of our linear ARPLN-ICA methodology in the revised manuscript, and establish the connection with non-linear ICA in the discussion surrounding Proposition 3.1.
>
> **"Lack of Quantitative Validation for Biological Interpretation ..."**
>
> Thank you for this insightful suggestion. We agree that the biological interpretation in the microbiome experiment was primarily qualitative. Since the authors of the microbiome dataset report symptomatic infection periods for each mouse but no additional clinical metadata, we added a quantitative validation based on the inferred source dynamics during commensal versus symptomatic phases following your suggestion.
>
> Specifically, for each mouse, we computed the difference between the average source value during the commensal phase ($t=8..13$) and during the symptomatic phase ($t=14..19$), yielding a source-shift statistic per mouse. Given the small number of samples, we view this evaluation as supportive rather than definitive, and report the average source shift and the fraction of total absolute shift explained by each source in the table below.
>
> This quantitative analysis complements the qualitative interpretation based on loadings and switch alignment, and will be included in the revised manuscript.
>
> |Source|Avg. shift|Avg. frac. shift|
> |---|---|---|
> |1|-0.16 (0.07)|10% (5%)|
> |2|+0.20 (0.17)|13% (7%)|
> |3|-0.19 (0.10)|10% (3%)|
> |4|-1.20 (0.44)|67% (3%)|
>
> **Table 2:** Source shifts in the microbiome experiment from commensal to symptomatic (with std).
>
> **About the runtime analysis and numerical optimization:**
>
> Thank you for this remark. Our implementation already supports mini-batching for faster optimization convergence, although the reported experiments were run in full batch for consistency. We note, however, that mini-batching is not expected to reduce per-epoch runtime here, but may benefit memory usage and ELBO maximization convergence speed. Regarding sparse approximations, our current variational family already exploits factorization across latent dimensions (see Eq. (49)).
>
> We also note that both the simulations in Figure 3 and the microbiome experiment in Figure 13 indicate stable ICA estimation in relatively small-sample regimes, where runtime is low. More generally, the linear ICA setting considered here is typically less data-hungry than neural non-linear ICA methods. At the same time, the current method uses a structured approximation chosen to preserve the temporal dependencies of the model, which comes at additional computational cost. Further scalability gains would likely require relaxing part of this structure through a different approximation, which we view as an interesting direction for future work.
>
> [1] Hälvä et al. "Disentangling identifiable features from noisy data with structured nonlinear ICA." NeurIPS (2021)

---

> > ### Author Rebuttal · Reviewer_sfAG · 2026-04-03
> >
> > The reviewer thanks the authors for their responses.

---

### Official Review · Reviewer_qTS7 · 2026-03-13

**Soundness:** 3
**Presentation:** 3
**Significance:** 2
**Originality:** 3
**Overall Recommendation:** 4
**Confidence:** 3

**Summary:**

This paper proposes an Independent Component Analysis (ICA) framework for temporal count data. Detailed theoretical analysis demonstrates the identifiability of the model. The proposed method learns the model parameters through an efficient amortized variational inference procedure. Experiments verify that the proposed method effectively captures the longitudinal dynamics.

**Compliance With Llm Reviewing Policy:**

Affirmed.

**Final Justification:**

The author clarifies in detail the applicability of the proposed theories and methods.

**Key Questions For Authors:**

See above.

**Limitations:**

See above.

**Strengths And Weaknesses:**

Strengths:
1. For time series count data and regime-adaptive dynamics scenarios, the ARPLN-ICA model is proposed. It can capture the statistical characteristics of such data more accurately than the traditional ICA model, thereby better identifying and adapting to the regime transitions in the data that change over time, and supporting principled interpretation.
2. The paper provides detailed theory and proof, demonstrating that the proposed ARPLN-ICA model is identifiable. And the experimental result demonstrates the effectiveness of the proposed methods.

Weaknesses:
1. This model assumes that the latent components satisfy a linear relationship within a specific mechanism. This may limit the model's modeling capabilities and make it unsuitable for complex nonlinear relationships in the real world. Therefore, it may not be well applied to real-world scenarios.
2. Although the effectiveness of the proposed method has been demonstrated in the context of microbiomes, extending the framework to other domains or more complex data types remains a non-trivial task, requiring targeted adjustments.

---

> ### Author Rebuttal · Authors · 2026-03-30
>
> Dear reviewer, we appreciate your positive and insightful review. Please find our responses below:
>
> **"This model assumes that the latent components satisfy a linear relationship within a specific mechanism. [...] it may not be well applied to real-world scenarios."**
>
> Thank you for this remark which allows us to clarify our contribution.
> We would like to insist on the fact that our approach is not limited to linear ICA. Although the latent mixing is linear, the sources are governed by a regime-switching mechanism yielding non-linearity, while the observation model Eq.(2) involves a linear transformation coupled with a non-linearity. This makes the overall proposed framework much more involved than traditional linear ICA, both from theoretical and practical perspectives.
>
> More broadly, our framework and theoretical results are compatible with non-linear mixing. Proposition 3.1 shows that identifiability in PLN-ICA reduces to the identification of the latent log-intensity process, which allows us to analyze non-linear PLN-ICA by combining our observation model with existing non-linear ICA identifiability results, such as Theorem 2 of [1].
> Our focus on the linear ICA setting is mainly motivated by the theoretical grounding this methodology provides, notably the mild guarantees under which the model becomes interpretable, and its simplicity, which is compatible with low-sample-size regimes such as our microbiome study, conversely to non-linear ICA that can be data-hungry. While we agree that exploring other non-linear ICA mixing mechanisms can increase the expressivity of the model, we believe our application to microbiome data displays the relevance of our approach in a complex real-world scenario.
>
> We propose a revision of the paper to clarify this point, and to add in Section 3.2 how our results can be used to derive non-linear identifiable PLN-ICA, following notably Theorem 2 of [1].
>
> **"Although the effectiveness of the proposed method has been demonstrated in the context of microbiomes, extending the framework to other domains or more complex data types remains a non-trivial task, requiring targeted adjustments."**
>
> We agree that demonstrating the method capabilities on additional real-life applications would further strengthen the empirical validation. However, we would like to clarify that applying ARPLN-ICA in other domains does not require methodological modifications of the underlying framework, or the inference procedure. In practice, applying our method mainly involves standard model selection choices, such as the latent dimension, the number of switching regimes, and the architecture of the amortized networks.
>
> To support applicability across domains, we conducted a preliminary analysis on the NOAA Storm Events database, a climate dataset counting severe weather events in the USA between 2000 and 2025. We considered 16 continental event types aggregated monthly and reorganized the data into annual trajectories to capture seasonal structure. We then applied the same pipeline as for microbiome data to illustrate the flexibility. The table below shows the LOO medoid mixing, consistent with known seasonal and meteorological patterns: C1 contrasts severe convective hazard with moisture-related events [2]; C2 contrasts excessive heat against winter hazards; C3 is a compound warm-season hazard; C4 contrasts synoptic winter hazards with heat-related convective ones.
>
> Additional interpretations are provided in our response to reviewer LkU2 and will be incorporated into the revised manuscript.
>
> |Event type|C1|C2|C3|C4|
> |---|---|---|---|---|
> |Cold/Wind|-0.39 (0.06)|-0.74 (0.03)|-0.13 (0.03)|+0.24 (0.02)|
> |Drought|-1.00 (0.04)|+0.17 (0.05)|+0.14 (0.01)|+0.06 (0.00)|
> |Excessive Heat|-0.26 (0.05)|+0.84 (0.05)|+0.82 (0.04)|-1.00 (0.05)|
> |Flash Flood|-0.40 (0.03)|+0.25 (0.01)|+0.48 (0.01)|-0.26 (0.02)|
> |Flood|-0.14 (0.02)|-0.13 (0.01)|+0.30 (0.01)|-0.07 (0.01)|
> |Hail|+1.00 (0.03)|-0.29 (0.04)|+1.00 (0.02)|-0.33 (0.02)|
> |Heavy Rain|-0.48 (0.03)|+0.17 (0.02)|+0.56 (0.01)|-0.23 (0.01)|
> |Heavy Snow|-0.36 (0.03)|-0.90 (0.03)|-0.19 (0.02)|+0.49 (0.02)|
> |High Wind|-0.41 (0.02)|-0.40 (0.03)|+0.04 (0.01)|+0.26 (0.01)|
> |Thunderstorm Wind|+0.35 (0.03)|+0.11 (0.02)|+0.53 (0.01)|-0.35 (0.02)|
> |Tornado|+0.23 (0.03)|-0.20 (0.02)|+0.71 (0.02)|-0.22 (0.01)|
> |Winter Storm|-0.24 (0.04)|-1.00 (0.03)|-0.23 (0.02)|+0.53 (0.03)|
> |Blizzard|-0.26 (0.04)|-0.95 (0.03)|+0.12 (0.03)|+0.40 (0.01)|
> |Dense Fog|-0.49 (0.02)|-0.36 (0.04)|+0.20 (0.02)|+0.10 (0.01)|
> |Frost/Freeze|-0.32 (0.02)|-0.34 (0.02)|+0.51 (0.02)|+0.05 (0.01)|
> |Lightning|-0.10 (0.05)|+0.35 (0.01)|+0.88 (0.01)|-0.64 (0.02)|
>
> **Table 1:** Medoid ICA mixing (relative to max per component; std across LOO in parentheses)
>
> [1] Hälvä et al. "Disentangling identifiable features from noisy data with structured nonlinear ICA." NeurIPS (2021)
>
> [2] Barton et al. "Soil moisture gradients strengthen mesoscale convective systems by increasing wind shear". Nature
> Geoscience (2025)

---

> > ### Author Rebuttal · Reviewer_qTS7 · 2026-04-03
> >
> > Thanks for your detailed response.

---

### Decision · Program_Chairs · 2026-04-30

**Decision:**

Accept (regular)

**Comment:**

This paper makes a solid methodological contribution to an important and under-studied problem: identifiable independent component analysis for temporal count data. The work combines regime-switching dynamics with Poisson log-normal emissions in a principled framework with formal identifiability guarantees and efficient variational inference.

The primary weakness is limited empirical validation as only one real-world dataset is presented in the main paper, and some coverage of prior work and relevant baselines is lacking. Authors partially addressed this during rebuttal with a preliminary analysis of NOAA climate data and new baselines (rslds, p-dlds, Gaussian observation models). The authors also revised the paper to include missing citations on Poisson GLM models. These revisions greatly improve the paper, though it should be noted that some reviewers remained reserved, saying that their concerns are simply "not easily addressed in a short rebuttal" and that the "limited evaluation" remains an issue.

On balance however, the methodological contributions are original and genuine. As one reviewer stated, the "integration of [Poisson log-normal] and switching linear dynamics is a principled and innovative combination, with no direct prior work unifying these components for ICA." All four reviewers ultimately supported acceptance, and all marked their major concerns as at least partially resolved after the rebuttal. The paper merits acceptance, with the understanding that the camera-ready should incorporate the additional baselines and climate experiments promised during discussion.